# Contingency and selection in mitochondrial genome dynamics

Christopher J Nunn[1]*, Sidhartha Goyal[1,2]*

[1]Department of Physics, University of Toronto, Toronto, Canada; [2]IBBME, University of Toronto, Toronto, Canada

**Abstract** High frequencies of mutant mitochondrial DNA (mtDNA) in human cells lead to cellular defects that are associated with aging and disease. Yet much remains to be understood about the dynamics of the generation of mutant mtDNAs and their relative replicative fitness that informs their fate within cells and tissues. To address this, we utilize long-read single-molecule sequencing to track mutational trajectories of mtDNA in the model organism *Saccharomyces cerevisiae*. This model has numerous advantages over mammalian systems due to its much larger mtDNA and ease of artificially competing mutant and wild-type mtDNA copies in cells. We show a previously unseen pattern that constrains subsequent excision events in mtDNA fragmentation in yeast. We also provide evidence for the generation of rare and contentious non-periodic mtDNA structures that lead to persistent diversity within individual cells. Finally, we show that measurements of relative fitness of mtDNA fit a phenomenological model that highlights important biophysical parameters governing mtDNA fitness. Altogether, our study provides techniques and insights into the dynamics of large structural changes in genomes that we show are applicable to more complex organisms like humans.

## Editor's evaluation

This work provides further insight into long-time outstanding questions in the field of mitochondrial genetics using long-read sequence analysis and biophysical modeling. Specifically, the authors show how replication origins in mitochondrial DNA are recombination hotspots that can result in excision cascades, that lead to a variety of different mitochondrial mutants and in some cases even heteroplasmic cells. Finally, crossing wild-type cells with various mitochondrial mutants allowed the development of a model for the suppressivity (fitness) of different mitochondrial variants that suggests that the density of replication origins in different repeated units is a major determinant of mtDNA suppressivity.

*For correspondence:
cnunn@physics.utoronto.ca
(CJN);
goyal@physics.utoronto.ca (SG)

**Competing interest:** The authors declare that no competing interests exist.

## Introduction

The mitochondrial DNA (mtDNA) in eukaryotic cells encodes a subset of enzymes involved in cellular respiration. Interestingly, the integrity of mtDNA has been implicated in critical biological processes other than respiration such as in apoptosis, trace element and intermediary metabolism, heme synthesis, and iron-sulfur cluster biogenesis (*Veatch et al., 2009*). Because mtDNA exists in multiple copies within numerous mitochondrial compartments, localized mtDNA damage produces heteroplasmic states with coexisting mutant and wild-type mtDNA in cells. Both intracellular mtDNA dynamics and intercellular selection then ultimately shape the fate of cell populations, with mtDNA damage resulting in cellular defects in single-celled organisms such as yeast and aging and disease in multicellular organisms such as humans.

In humans, large mtDNA deletions accumulate during the course of aging in skeletal muscle and brain tissue (*Fayet et al., 2002*; *Kraytsberg et al., 2006*; *Payne and Chinnery, 2015*), and result in

observable cellular defects (**Chan, 2006**). The same types of deletions have also been implicated in numerous diseases, including Parkinson's disease (**Bender et al., 2006**). While these mtDNA deletions have been widely observed, much remains to be understood about the dynamics that lead to the propagation of these deletions within cells and resultantly in the tissues of humans. This is partly because of the strong dependence of human cells on mtDNA for their survival, and the complexities in artificially creating heteroplasmy or modifying mtDNA in mammalian systems (**Mok et al., 2020**). To address some outstanding questions regarding the dynamics of mutant mtDNA, in this article, we explore mtDNA dynamics using yeast as a model organism. Yeast is particularly well suited to study mtDNA dynamics due to the dispensability of mtDNA and because heteroplasmic cells containing mutant and wild-type mtDNA are easy to artificially construct.

In yeast, mtDNA deletions were first linked to the Petite phenotype (**Ephrussi, 1949**; **Ephrussi, 1953**; **Ephrussi et al., 1949**). Petite colonies are smaller than their wild-type counterparts under respiration conditions due to mtDNA deletions that render cells incapable of respiration. These deletions are due to destructive recombination events between short repeated homology in mtDNA that excise portions of the wild-type genome (**Bernardi et al., 1976**; **Bernardi and Bernardi, 1980**; **Marotta et al., 1982**; **de Zamaroczy et al., 1983**). Excision events are followed by selection for sub-genomic (nonfunctional) mtDNA fragments that contain a high density of replication origins (**Goursot et al., 1980**; **Blanc and Dujon, 1980**; **de Zamaroczy et al., 1979**; **de Zamaroczy et al., 1981**). When subgenomic fragments have replication origin densities higher than the larger wild-type genome, they consistently outcompete or 'suppress' wild-type genomes within cells. Suppressivity, which is a measure of this replicative advantage of Petite mtDNAs over wild-type, was shown to correlate with origin density and was reduced when replication origins were disrupted or absent, constituting the rules of suppressivity (**de Zamaroczy et al., 1981**; **Mangin et al., 1983**; **Bernardi, 2005**). Altogether, rolling circle replication coupled with this excision and selection for replication origins results in the formation of complex concatemer structures in the mtDNA of Petite cells that often contain multiple replication origins from distant locations of the wild-type genome (**Locker et al., 1974**; **Locker et al., 1979**; **Faugeron-Fonty et al., 1979**).

Near the time of the complete sequencing of the mitochondrial genome of yeast in 1998 (**Foury et al., 1998**), work on the structural details of mtDNA that lead to the aforementioned discoveries in Petites appeared to wane. However, a number of open questions about the dynamics of Petite mtDNAs, which are at their core a result of mtDNA deletions, remain to be explored fully.

Regarding the structure and generation of Petite mtDNA, three questions remained to be addressed. These include: (1) *What drives mtDNA excision events in Petites to cluster near replication origins?* Previous work shows that excisions occur all throughout the genome but with a higher density near replication origins (**Bernardi and Bernardi, 1980**; **de Zamaroczy et al., 1983**; **de Zamaroczy and Bernardi, 1986**; **Marotta et al., 1982**; **Osman et al., 2015**). The interplay between location-specific excision frequencies and selection for origin-containing fragments remains entangled. (2) *What is the nature and dynamics of the ongoing excision cascades in Petites, that is, how do subsequent excisions relate to previous excisions?* Previous work in **Bernardi et al., 1976**; **Lewin et al., 1978**; **Lewin et al., 1979**; **Locker et al., 1979** showed qualitatively that persistent heterogeneity in mtDNA content was present in the sequencing of Petite strains, pointing to continuing excision events. However, these works did not provide a quantitative description of this heterogeneity or explore the relationship between the structure of the coexisting mtDNAs. (3) *Are the contentious and rare non-periodic mtDNA structures observed in yeast real?* The studies of **Heyting et al., 1979**; **Bos et al., 1980** provides evidence for non-periodic structures, which is unexpected given that rolling circle replication produces periodic, tandemly duplicated structures. The work in **Faugeron-Fonty et al., 1983** refutes these observations, providing a conflicting hypothesis which remains to be reconciled.

Concerning the distribution of mtDNAs at a cell population level, another open question is: (4) *How is the observed structural heterogeneity of mtDNA in yeast colonies partitioned among individual cells?* The work in **Lewin et al., 1978**; **Lewin et al., 1979**; **Locker et al., 1979** points to homoplasmic contributions to Petite-colony heterogeneity. The extent of homoplasmic and heteroplasmic contributions to colony-level heterogeneity remains to be quantified.

And finally, given an understanding of both mtDNA structure dynamics and its partitioning in populations, the final question we address is related to the structure-function relationship of mtDNA: (5) *What contributes to the fitness of mtDNA structures, and how does structure inform suppressivity?*

The suppressivity rules provided previously in *de Zamaroczy et al., 1981*, which described how mtDNA structure influenced suppressivity, were limited to reduced Petite genomes with small sizes and relatively high suppressivities. Do these same rules explain suppressivity across a larger range of genome structures and suppressivities, and can we construct a biophysical model of suppressivity that relies on these rules?

In this study, we address each of these long-standing questions with new long-read sequencing technology and accompanying structural inference methods. We highlight some advantages of Nanopore sequencing in addressing these questions and future ones, but also technical challenges specific to structure reconstruction with Nanopore sequencing of the mtDNA in yeast. Among new answers to all of the aforementioned questions, we showcase a previously unseen pattern that constrains subsequent excision events in generating new Petite mtDNA structures from existing ones, settle contention in the literature surrounding the existence and generation of non-periodic 'mixed' Petite structures, and propose a phenomenological model of suppressivity that highlights important biophysical parameters governing mtDNA fitness. Finally, we connect these observations in yeast to mtDNA deletions in humans which exhibit remarkably similar patterns.

## Results
### Overview of the structure of Grande and Petite mtDNA

To quantify mtDNA structure and their dynamics, we opted to sequence both Petite and Grande colonies with a Nanopore MinION single-molecule sequencing platform. We expected the long reads generated from this sequencing technology to improve structure reconstruction for both high and low frequency structures compared to short-read sequencing approaches. In total, we sequenced 38 Petite colonies derived from 9 spontaneous Petite colonies through passaging and 10 Grande (wild-type) colonies of the same *Saccharomyces cerevisiae* strain. Four of these Grande colonies were cultured under non-fermentable media (YPG), and six in fermentable media (YPAD). Starting with nine spontaneous Petite colonies, each colony was passaged twice onto new media (YPAD), storing and culturing three colonies at each passage. This generated families of Petite colonies, with nine colonies sharing each spontaneous progenitor after two passages (*Figure 1a*). The suppressivities of all colonies sharing a progenitor were measured (see Materials and methods), but only a subset was sequenced (dotted circles in *Figure 1a*). Subfamilies, labeled in *Figure 1a* as a subscript, were grouped based on differing mtDNA content from other members in the same family. The coverage curves from the sequencing of each Petite colony and a subset of Grande colonies are shown in *Figure 1b* and provide a coarse picture of their mtDNA content. It is evident that some Petite colonies within families, such as families 1 and 4, have differing mtDNA content but share the same spontaneous Petite colony progenitor. This diversity is most likely the result of early mtDNA instability in spontaneous progenitor colonies that segregate into different cells through genome bottlenecks and are sampled through passaging. It is also possible that this diversity is a result of ongoing mtDNA changes during the growth of the colony before sequencing. Nevertheless, comparing the mtDNA content in colonies that share second passage progenitors reveals that two passages followed by culturing (~32 generations) was sufficient to homogenize mtDNA content in all cases except family 1. Family 1 we believe to be a special case where early mtDNA instability occurred in the spontaneous progenitor, and then again in the second passage progenitor of colonies 1b and 1c or during their growth. Given that we sampled such a case in our experiments suggests roughly 1 in 10 chances of such events, but would require a larger study to quantify it.

Mapping of the mtDNA to a reference sequence, followed by careful filtering of inverted duplication artifacts (*Appendix 1—figure 1*) and clustering of alignment breakpoint signals with a variety of parameters (see Materials and methods), revealed both inverted and non-inverted mtDNA breakpoints in all Petite colonies and rare mtDNA breakpoints in Grande colonies. These breakpoint signals delineate sequence alignments that are collinear with the reference mtDNA sequence, but merged in such a way that disjoint alignment locations on the reference genome have been brought together. Non-inverted breakpoints indicate the merging of disjoint sequences in the reference from the same strand, or with the same orientation, while inverted breakpoints indicate the merging of disjoint mtDNA sequences on opposite strands (*Figure 2a*). Long reads with an average read length of 6 kbp and maximum length of 120 kbp directly revealed that these breakpoint signals were contributed by

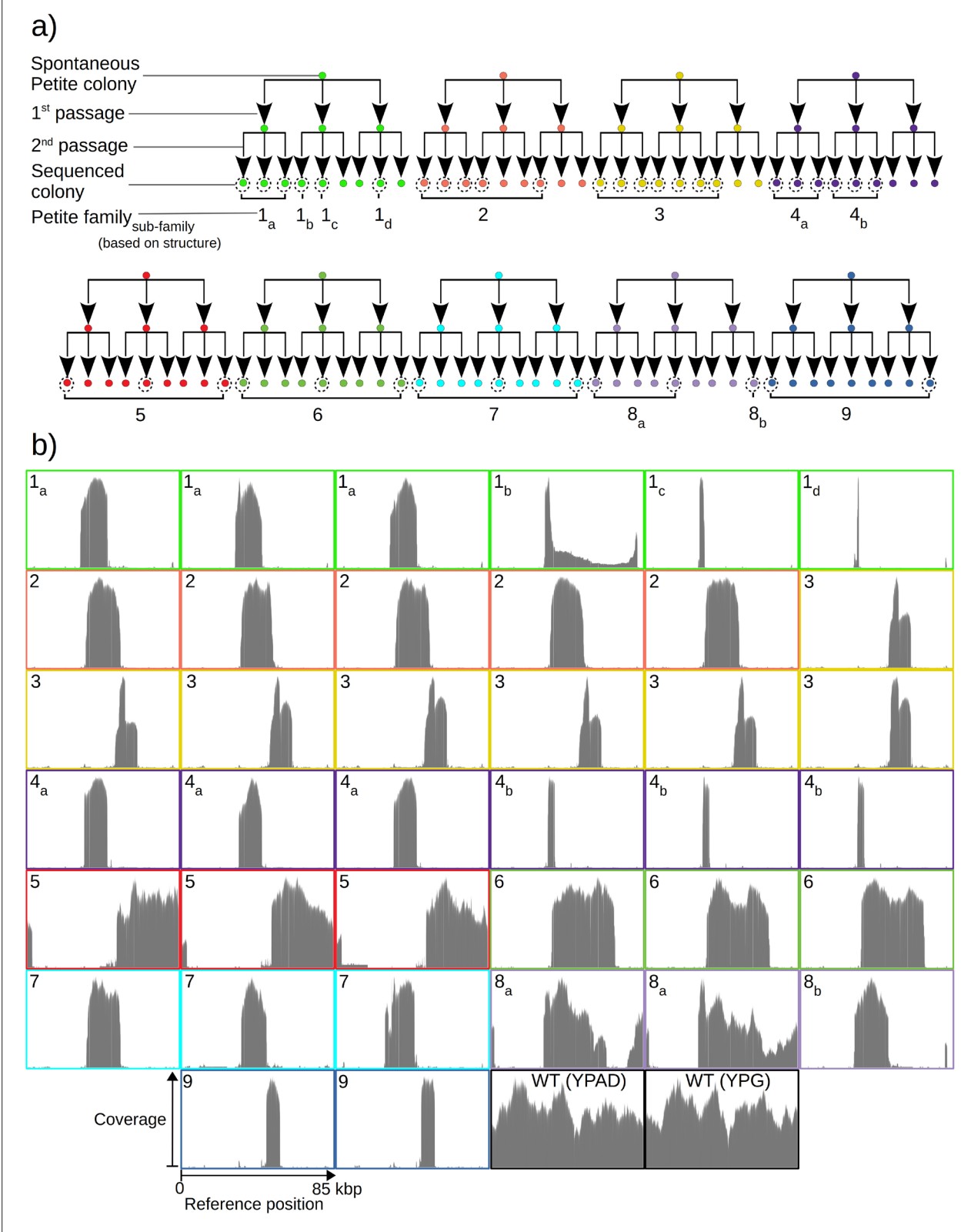

**Figure 1.** Overview of the experiment and observed mtDNA diversity in sequenced yeast colonies. (**a**) An overview of the architecture of the Petite colony sequencing experiment in this study. Nine spontaneous Petite colonies were passaged twice onto new media, culturing and storing three colonies for each passage. This produced families of colonies (indicated by color), where all colonies after two passages were derived from the same spontaneous Petite colony progenitor, but only a subset of colonies was cultured and then sequenced with a Nanopore MinION sequencing device

*Figure 1 continued on next page*

*Figure 1 continued*

(indicated by dotted circles). In addition to families, subfamilies are labeled as a subscript and grouped based on the predominant mtDNA structure present in these colonies according to sequencing results. (**b**) The sequencing coverage (arbitrary coverage scaling, consistent genome reference location) in all Petite colonies in addition to a subset of wild-type (WT), or Grande colonies sequenced. Ten Grande colonies were sequenced as a reference, four after growth under non-fermentable conditions (YPG), and six under fermentable conditions (YPAD). Border colors correspond to (**a**), and black borders are examples of Grande colony coverages.

concatemer structures in Petites, composed of tandem repeats of sub-genome sized repeat units that had been excised from the wild-type genome and amplified into repeated structures (*Figure 2b* — leftmost spiral). Grande reads also revealed concatemer structures manifested in reads as subsamples of genome-sized repeating units devoid of breakpoint signals (*Figure 2b* —rightmost spiral). These concatemer structures in Petites with sub-genome sized repeat units are consistent with the existing literature on mtDNA structure in Petites that relied on restriction digestion mapping and electron microscopy (*Bernardi et al., 1975*; *Bernardi et al., 1976*; *Lewin et al., 1978*; *Locker et al., 1979*; *Faugeron-Fonty et al., 1979*; *Bernardi and Bernardi, 1980*).

All Petite colonies contained at least one breakpoint signal and often a diverse set of breakpoints totaling 84 breakpoints across 38 Petite colonies, whereas in Grande colonies sequenced only 2 had high confidence breakpoints detected, with a total of 3 breakpoints across 10 Grande colonies (*Figure 2c* and *Appendix 1—figure 2*). The diversity in location of mtDNA breakpoints within Petite families and breakpoint counts greater than the number of members of each family/subfamily also echo the diversity observed in the coverage plots but with more detail. These diverse breakpoint distributions within families indicate either structural diversity in the progenitor colony, continued changes in mtDNA structure resulting in subfamilies or coexisting structures in colonies, or multiple breakpoint signals within colonies indicating more complex mtDNA structures generated by multiple excision events.

## What drives mtDNA excision events in Petites to cluster near replication origins?

The prevailing theory for the formation of Petites relies on sequence-specific illegitimate recombination within the wild-type DNA molecule between repeated GC clusters and AT stretches (*Bernardi and Bernardi, 1980*; *de Zamaroczy et al., 1983*; *de Zamaroczy and Bernardi, 1986*), which are prevalent in all noncoding regions of the mitochondrial genome in yeast. In particular, the extensive homology of the eight mitochondrial origins of replication and their inclusion of similar GC clusters (*de Zamaroczy and Bernardi, 1986*) suggest important regions for illegitimate recombinations. Evidence for hybrid origins resulting from recombination between adjacent origins in the wild-type genome have been seen in restriction digestion data (*Marotta et al., 1982*). Large structural variations and smaller mtDNA variations have also been observed in Illumina sequencing of Petites to cluster within origins and within close proximity to origins (*Osman et al., 2015*). Given that replication origins were implicated in previously observed Petite mtDNA excisions and a variety of mtDNA variations, we were curious to understand the involvement of replication origins across the diverse set of excision events we observed. Placing non-inverted breakpoint locations and replication origin locations on mitochondrial reference coordinates reveals the clustering of breakpoints near edges of interacting origins of the same orientation (*Figure 3a*). In fact, ~30% of breakpoints reside within replication origins, indicating that the structures containing these breakpoints have origins that are perturbed by excisions, or hybrid origins. An additional ~20% of breakpoints reside within 275 bp of the edge of an origin. The remaining 50% of breakpoints are located from 275 bp to 3 kbp from the edge of an origin.

Next, we asked if this non-uniform pattern of excision revealed new rules for mtDNA fragmentation. Besides the potential role of homology of the replication origins, it has been noted that a high density of unperturbed replication origins in Petite structures result in a replication advantage for Petite mtDNAs over wild-type mtDNAs (*Blanc and Dujon, 1980*; *de Zamaroczy et al., 1981*; *Mangin et al., 1983*). In concatemer structures, this means that smaller repeated fragments containing replication origins are more fit than wild-type fragments when in competition with each other. This leads to a natural question of whether or not clustering near origins is due to higher frequency recombination within or near replication origins, or if it is due to selection on a pool of arbitrary excisions with selection for the resulting small fragments containing replication origins. To this end, we compared the

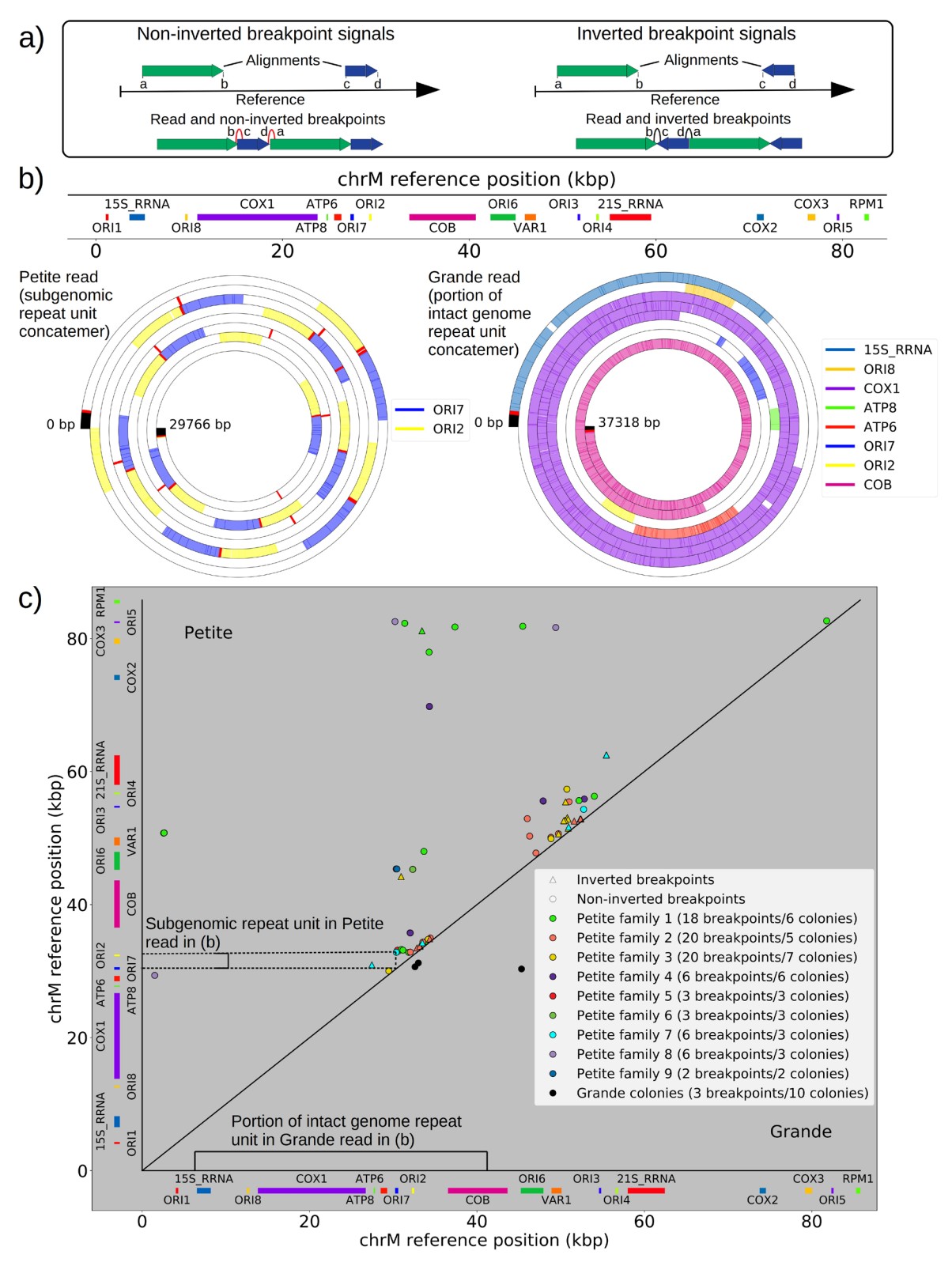

**Figure 2.** The mtDNA in yeast exists as concatemers that are delineated by breakpoint signals in sequencing alignments. (**a**) A schematic of the definition of alignments and breakpoint signals. Alignments (which are sequences collinear with the reference genome) with their location on the reference are shown as colored arrows alongside the coordinates of alignment edges (a, b) and (c, d). A hypothetical read is shown below these alignments, indicating how the alignments are oriented with respect to each other and the coordinates of the alignment edges in contact that define

*Figure 2 continued on next page*

Figure 2 continued

the breakpoints denoted as arcs. Non-inverted breakpoints represent merged alignments from disjoint locations on the reference that map to the same strand of DNA (red arcs), while inverted breakpoints represent the same disjoint merging of alignments but with different orientation (black arcs). (**b**) Representative examples of mtDNA structures from sequencing reads in Petite and Grande samples. The top of this panel shows the mitochondrial reference and annotated features of the genome in colored blocks. Below the reference are long sequencing reads wrapped around themselves in a spiral that display the same annotated features as colored blocks. These spiral plots also include red bars which indicate breakpoint locations. Black regions indicate unmapped portions near the ends of the read due to adapters and barcodes. The spiral on the left is a sequencing read from a Petite colony, showing that two origins of replication have been excised from the wild-type genome and tandemly repeated in a concatemer structure. The spiral on the right is a read from a Grande colony showing a portion of a linear segment of the genome, without breakpoints (red bars) except at the ends of the reads which mark the end of alignments. (**c**) Summary of mtDNA breakpoints detected across 38 Petite colonies, that were derived from 9 spontaneous petite colonies through passaging (above diagonal), and 10 Grande colonies (below diagonal). In this scatter plot, each marker represents the centroid of a cluster of mtDNA breakpoint signals in reference coordinates from reads in a single sample. X and Y coordinates of each marker are the regions of mtDNA that interact to produce the breakpoint. The numbers of breakpoints in each family are indicated in the legend, as well as the numbers of colonies in each family sequenced. Also indicated are the regions on the reference genome that make up the repeating unit in the Petite read shown in (**b**) and the subsection of the reference genome that contributes to the Grande read in (**b**).

distribution of displacements between breakpoints and the closest replication origins (*Figure 3b*, blue curve) to three different models. In the first model (*Figure 3b*, orange curve), we plotted the same displacement distribution for uniform random mtDNA fragments, with a size distribution given by the sequencing data, placed on the reference mitochondrial genome. In the second model (*Figure 3b*, black curve), we plotted the displacement distribution for random fragments between perfect repeats of greater than 11 bp in the reference, which is motivated by the fact that excisions require perfect repeats or highly homologous regions. In the third model (*Figure 3b*, green curve), we plotted uniform random fragments as in the orange curve, with a length distribution from the sequencing data, but conditioned on spanning a portion of a randomly selected origin of replication. A schematic summarizing these models is provided in *Figure 3c*.

The orange and black distribution captures the breakpoint displacements from origins expected from random excision events and no selection for replication origin containing fragments. Note that the similarity between the black and orange curves demonstrates the prevalence of repeated homology in the mitochondrial reference genome. The green curve captures random excision, but strong selection for small replication origin containing fragments due to the requirement for alignments to contain a portion of an origin in this model. The observed data agrees nicely with the green curve for most of the domain of the distribution. The empirical origin to breakpoint distributions and their agreement with the green model are also consistent across a variety of breakpoint clustering/filtering parameter regimes, which have minor effects on the individual breakpoints extracted from the sequencing data, but little effect on these distributions (*Appendix 1—figure 3*, *Appendix 2—table 2*). Thus, while we cannot exclude a model of non-random excisions favoring close origin proximity, the bulk of the minimum breakpoint to origin displacement distributions observed can be explained by random excision and strong selection for small origin containing fragments which is in agreement with the prevailing theory of Petite mtDNA formation.

## What is the nature and dynamics of the ongoing excision cascades in Petites?

Next, we were curious to know if we could observe ongoing excision cascades and whether or not subsequent excisions in Petites differed from initial excisions in Grandes that generated the first Petite mtDNAs. In 16 Petite samples sequenced from families {$1_a$, $1_b$, 2, 3, $8_a$} there were detectable levels of repeated structures that differed from the 'primary' mtDNA structures which span the longest portion of the reference genome and generally contribute to the majority of mtDNA (see Materials and methods on details of structure reconstruction). These lower frequency structures, or 'alternate' structures as they will be described from here on, were found to contribute from 0.1% to 59% of total mitochondrial content in these samples. Following multiple passagings of Petite colonies before sequencing, which would rapidly dilute any initially coexisting structures due to mtDNA bottlenecks (*Ling and Shibata, 2004*), these alternate structures most likely result from subsequent excisions of the primary structure during culturing. Such 'excision cascades,' where further excisions act on existing Petite fragments were hypothesized and discussed by *Locker et al., 1979*; *Marotta et al., 1982*; *Bernardi, 2005*, where it was suggested that the varying levels of alternate structures will depend

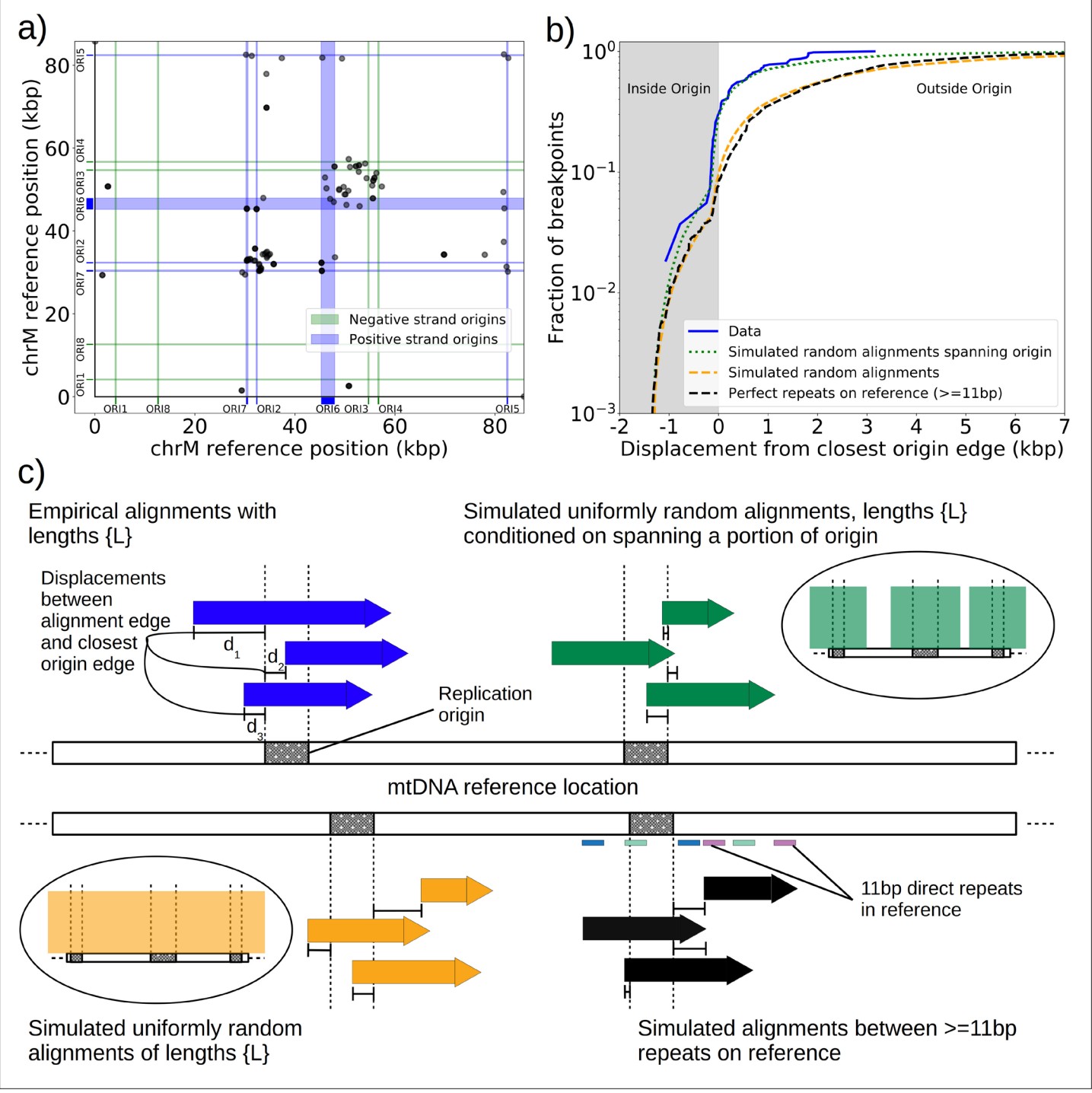

**Figure 3.** Replication origin and mtDNA excision proximity explained by random excision and selection for origin-dense fragments. (**a**) The colocalization of replication origins and alignment breakpoint locations due to excisions. Black dots represent the centroids of breakpoint clusters (see Materials and methods), and blue and green shading highlights replication origins and their orientation. Darker black dots are due to overlaps of breakpoints, indicating high densities of breakpoints at these locations. (**b**) A cumulative plot of the displacements between breakpoint edges and closest origins of replication, where the blue curve shows this enrichment of breakpoints near replication origins (top left, (**c**)). The orange curve represents a simulation of uniform random alignments placed on the reference genome following the true alignment length distribution in the data (bottom left, (**c**)). The black curve represents the simulation of alignments between randomly selected perfect repeats ≥11 bp on the reference sequence (bottom right, (**c**)). The green curve agrees much better with the data (blue) curve, which is the same simulation of random alignments placed on the reference following the length distribution of the data, but with the requirement that these alignments span some portion of a randomly selected origin of replication (top right, (**c**)). (**c**) A schematic of the models plotted in (**b**). Alignments are denoted as arrows, with distances between alignment edges

*Figure 3 continued on next page*

Figure 3 continued

(breakpoints) and replication origins (dotted boxes) as dimension lines. Circled drawings depict that uniformly random alignments are selected in the orange model, whereas alignments conditioned on spanning replication origins are present in the green model.

on their generation rate and selective advantage in replication over the primary structure. Part of what makes the dynamics of mtDNA within these cascades interesting is the multiplicity afforded by a large variety and number of potential excisions. Particular excisions that bring regions of homology together may open up entirely new trajectories of excision dynamics that were previously unlikely or inaccessible due to mtDNA conformation.

An extreme example of such an excision cascade is given in *Figure 4a*, where both primary and alternate alignments are shown on the reference mitochondrial genome location in addition to sequencing coverage. In *Figure 4b*, the structures of the repeated units composed of these alignments are also shown alongside their calculated mitochondrial content frequencies (see Materials and methods—Primary/alternate structural frequency calculations). The first thing to note is that the locations of alignments extracted from our structural detection pipeline that are involved in repeat units align well with the total sequencing coverage in *Figure 4a*. This diversity of alignments in *Figure 4a* is also corroborated by colonies that share the same progenitor. Each of the coverage curves of members of family 1 in *Figure 1b* share peaks with the coverage curve in *Figure 4a*. Second, there is significant diversity in the type of repeat units and necessary steps in their generation which stitch together these alignments in *Figure 4b*: Repeat unit #1 shown in *Figure 4b* (yellow and pink) is an example of a secondary excision across a segment of mtDNA containing the preexisting primary breakpoint, as both alignments come from opposite ends of the primary alignment and are stitched together. This immediately suggests that the excision occurred in mtDNA in a concatemer form, and across the repeat unit breakpoint (green and purple Type II excision; *Figure 4c*). Repeat unit #2 (blue, maroon, and green) also spans the primary breakpoint, but has an additional alignment in a different orientation that either resulted from two excisions or recombination of different repeat units (green, purple, and gray Type II excision; *Figure 4c*). Repeat unit #3 (green) is an example of an excision within the primary repeat unit and away from its edges (Type I excision; *Figure 4c*). Repeat unit #4 (purple) is an example of a repeat unit that shares only one edge with the primary alignment, but with this edge interacting with a different region of the genome producing a new alignment (Type III excision; *Figure 4c*).

In *Figure 4c*, we provide a schematic of the types of alternate repeat units observed across all samples and the plausible mechanisms of generation. We classify the resulting alternate repeat unit into three easily distinguishable classes in our data: Type I alternate repeat units are regions excised from the interior of primary repeat units. Type II alternate repeats contain or span the primary breakpoint, resulting from an excision across the breakpoint between two primary repeat units in a concatemer form. Type III alternate repeat units share one edge with the primary breakpoint and have a new edge within the primary alignment. For the technical details in the classification of these repeat unit types, see Materials and methods—Type I/II/III repeat unit classification. The proportions of each class of alternate repeats (35 total) across all samples are shown in *Figure 4d*, where it is clear Type III breakpoints make up the majority (57%) of alternate repeated structures observed across the 16 colonies where we see alternate structures. In *Figure 4e*, we also plot the distance between the closest primary and alignment edges for each class of repeat normalized by the primary alignment length. It is clear from this figure how close subsequent excisions are to primary breakpoints in the most abundant class, Type III, with a mean and median fractional distance between the alternate and primary edge of 2% and 1%, respectively. The mean and median fractional distances of Type II and Type III repeats are also comparable, which is only expected if Type III repeats truly share an edge of the primary breakpoint; Type II repeats directly recapitulate both primary alignment edges as they contain a perfect copy of the breakpoint. Meanwhile, Type III structures just need to have an edge close enough to either edge in the primary breakpoint to be within the sequencing error that defines the size of breakpoint clusters. So, the fact that both Type II and Type III are comparable in these distances, strongly suggests that Type III structures reuse part of the primary excision site. Therefore, the abundance of Type III repeats indicates a strong preference for secondary excisions at the site that produced the primary alignment itself, that ultimately constrains the trajectories of subsequent excision events in an unexpected, and previously unreported way. It is unclear at this time whether Type II

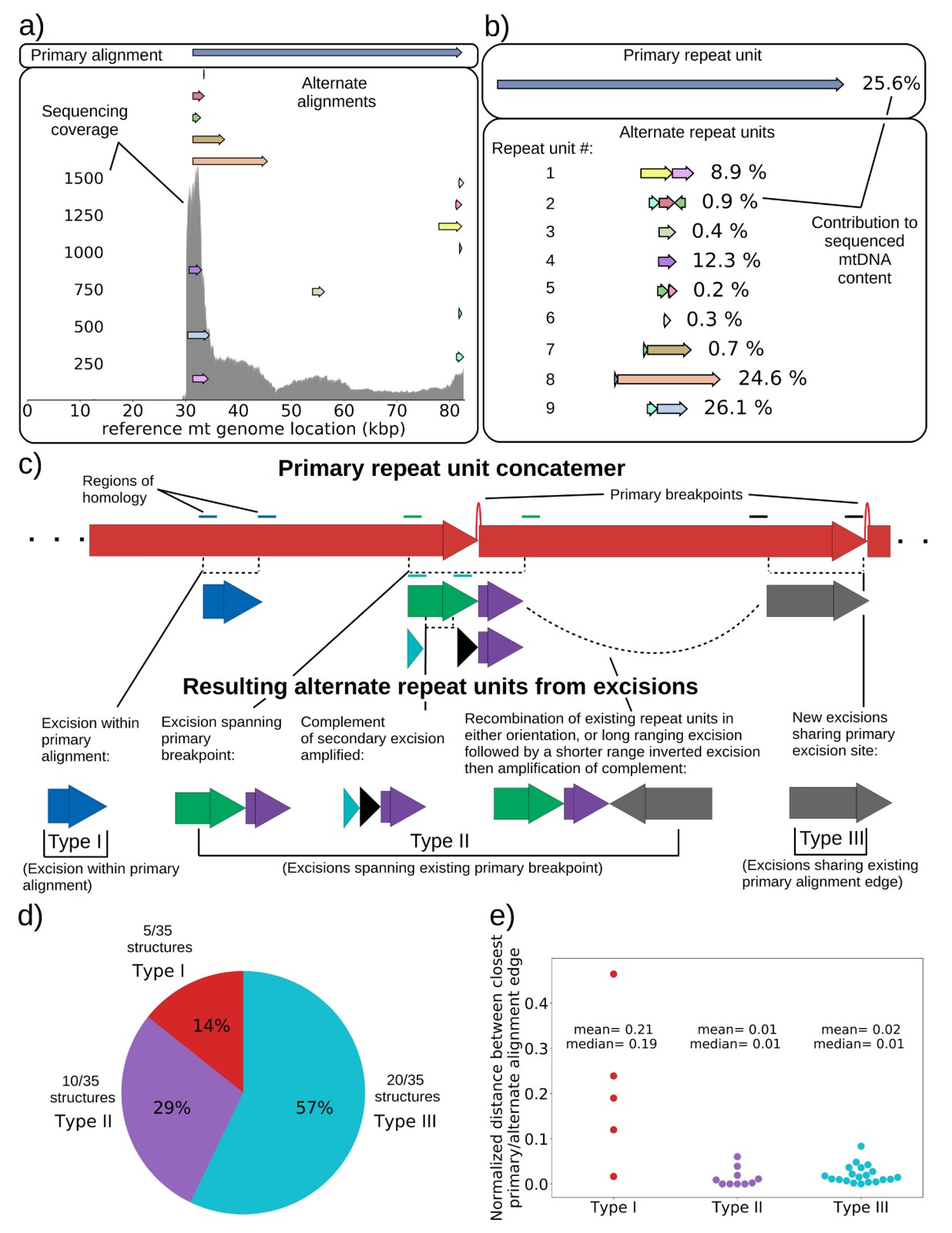

**Figure 4.** mtDNA excision cascades, quantification of colony structural diversity, and contingency in subsequent excisions. (**a**) Example locations of alignments from mapped reads (linear alignments here are bounded by breakpoints) observed in long reads within Petite sample 1_b. Endpoints of arrows are the mean breakpoint location for the cluster of breakpoint signals that punctuate alignments in repeats. The first panel shows a primary alignment which has the longest span on the genome and exists in long repeats. The second panel shows smaller alternate alignments that exist

*Figure 4 continued on next page*

*Figure 4 continued*

within detected repeated structures at a lower frequency. Also included in gray is a sequencing coverage map of this sample. (**b**) Excision cascade in Petite sample 1$_b$. This plot shows the same primary repeat unit in the first panel, and its contribution to total mitochondrial content as a percentage. The second panel shows the forms of alternate structures present in the same sample which were derived from the primary alignment, alongside their mitochondrial contribution as a percentage. (**c**) A schematic of the multiplicity of excision events that generate alternate repeat units. The primary concatemer is the red structure, where arrows indicate the alignments (contiguous regions of the reference) that make up the repeating units. The primary breakpoint between these alignments is denoted as a red arc. Colored rectangles above the alignments with the same colors indicate regions of homology in the primary structure that can interact to produce an excision. Dotted rectangles indicate excision sites that produce the alignments shown below them. In the lower half of the figure, five distinct excision events that can generate different repeat units are shown. These are grouped into three excision classes in the data: Type I, where excision occurs within primary alignments, Type II, where excisions span the existing primary breakpoint, and Type III, where excisions share one edge of the primary breakpoint. (**d**) The frequency of each class of excision across 35 alternate structures detected in the data. (**e**) A plot of the distance between alternate alignment edges and their closest primary alignment edge across all three classes, normalized by primary alignment length.

repeats, which encompass intact primary breakpoints, are related to this phenomenon. In general, this pattern of a preference of excisions around or across primary breakpoints is consistent across a variety of clustering/filtering parameters for breakpoints detection (*Appendix 1—figure 4*, *Appendix 2—Tables 1 and 2*), the details of which are described in methods.

## Are the contentious and rare non-periodic mtDNA structures observed in yeast real?

Seven colonies within family 3 in *Figure 1b* displayed distinguishably higher variance in coverage than the rest of the Petite colonies sequenced. This variance in coverage suggested either a complex repeat unit which itself contained smaller repeated units, or heterogeneity of mtDNA content in these samples. As such, we were interested to understand the source of this coverage variability. In these colonies, sequencing revealed non-periodic or non-tandemly duplicated primary structures involving partial inverted duplications of sequences. This is in contrast to the repeated units as concatemers that are found in the remainder of the Petite colonies and are primarily in tandemly repeated (non-inverted) forms. These non-periodic structures resemble the 'mixed' structures first characterized in detail in *Heyting et al., 1979*. The structure of one of these colonies is detailed in *Figure 5* and is representative of all seven 'mixed' structure colonies as they contain indistinguishable alignments and are derived from the same spontaneous Petite colony. In *Figure 5a*, four alignments of different lengths and reference locations are depicted as arrows. Note that all four alignments share the common region of the red alignment. Also included is the sequencing coverage of this particular colony, which aligns nicely with these alignments extracted from our structural repeat detection pipeline, as well as the ranked length and absolute count of alignments. Correcting for sampling bias (see Materials and methods—Mixed structure alignment frequency calculations) due to the sampled read length distributions across seven colonies with this same structure reveals that each alignment exists in equal proportions in colonies that harbor this structure (*Figure 5b*). Example structures in long reads selected from one of these samples are provided in *Figure 5c*, where the mixed structure is evident with seemingly random orientations of alignments. These 'mixed' structures are clear examples of intramolecular heterogeneity in mtDNA and likely intermolecular heterogeneity across a population of mtDNA fragments within cells given the differences in the content of the fragments observed. To attempt to make sense of this structure, which is at odds with the concatemer structures observed in all other colonies, we applied the repeat detection pipeline to see if any reads exhibited repeated structures. Interestingly, while superficially, this structure seems devoid of a clear pattern and appears uniformly randomized, across all seven colonies with the structure shown in *Figure 5c*, we did see some evidence of partially repeated structures, where the same alignments were repeated with the same orientation but separated by other single inverted alignments (*Figure 5d*). In all seven colonies with these same four alignments, these partially repeated structures are composed of the concatenation of largest and smallest alignments with opposite orientation, or the concatenation of the second-largest and second-smallest alignments.

The detection of the partial repeats in *Figure 5d* and the overlapping context in *Figure 5a* led us to the proposed mechanism of generation provided in *Figure 5e–f*. With this new evidence of partially detected repeat units, we build upon the crossover mechanism first hypothesized in *Bos et al., 1980*

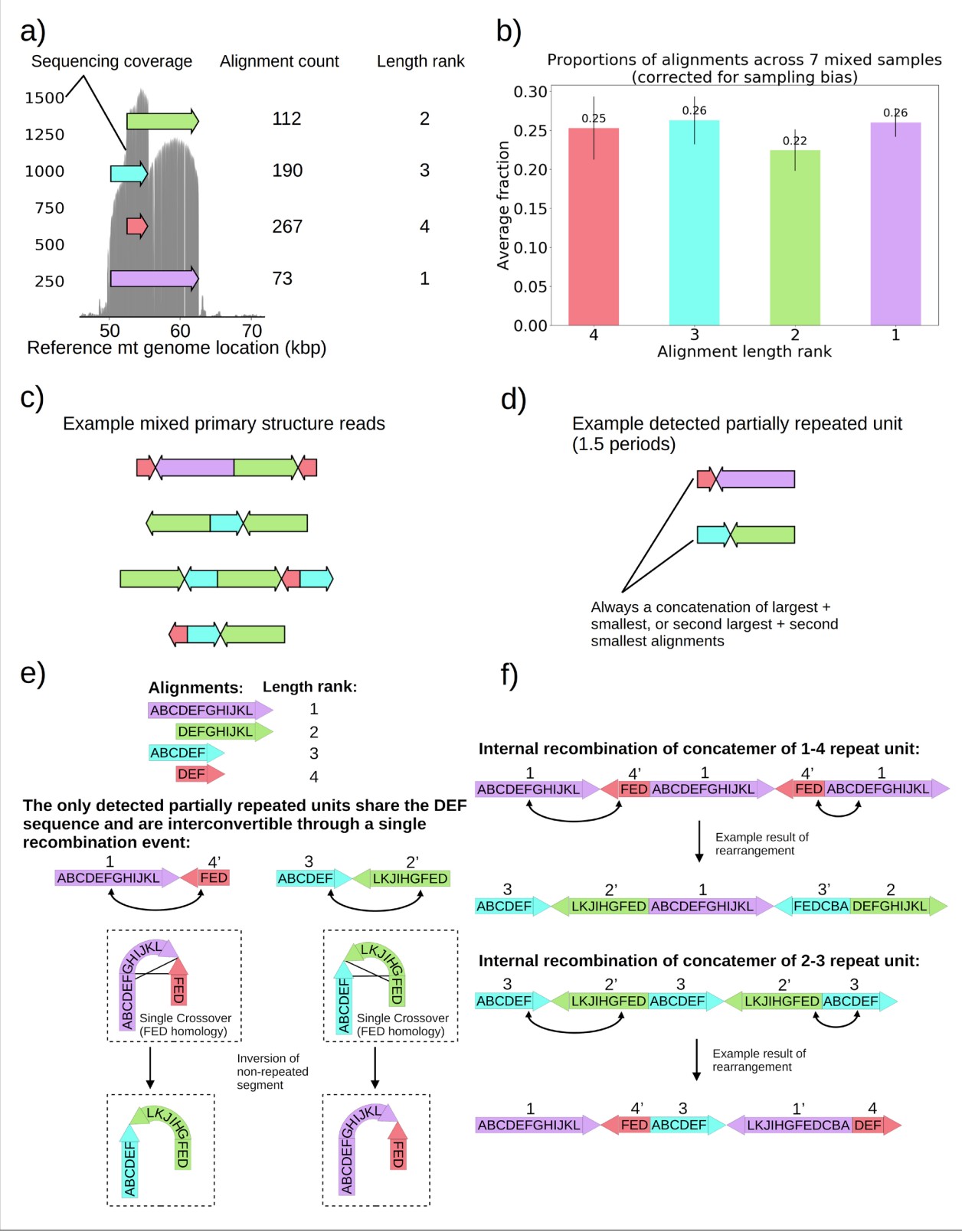

**Figure 5.** Non-periodic 'mixed' mtDNA structures and their mechanism of generation. (**a**) Alignment locations and their raw counts present in the primary structure of a 'mixed' repeat Petite colony. (**b**) Proportions of each four alignments across seven mixed Petite colonies sequenced after accounting for sampling bias (see Materials and methods). (**c**) Example structures in sequencing reads in this colony, displaying a collection of coexisting isoforms with identical base pair content but varying structures with two distinct inverted duplication breakpoints delimiting alignments. (**d**) Example

*Figure 5 continued on next page*

*Figure 5 continued*
partially repeated units detected after observing 1.5 periods in the repeat detection pipeline. (**e**) Interconvertibility of detected partial repeats. Arrow directions indicate the strand to which alignments have been mapped, in addition to the prime notation on length ranks of each structure which indicates an inverted alignment. (**f**) Crossover events in the background of concatemers that can produce all breakpoint transitions and structures observed in the data.

for what we believe is the same structure we observed. In *Figure 5e*, we show how the only detected partially repeat units are interconvertible through a single crossover event relying on interactions of oppositely oriented regions. This crossover event in concatemers or repeat units results in the inversion of the non-repeated sequences, and such an event can produce all of the 'mixed' read structures we observed across seven samples (*Figure 5f*). The generative picture given this proposed mechanism is the following: (1) First a recombination event produces one of the repeat units in *Figure 5d*, by a crossover mechanism like that suggested in *Faugeron-Fonty et al., 1983*, or origin-dependent mechanisms like those observed in yeast and proposed in *Brewer et al., 2011*; *Brewer et al., 2015*. (2) This repeat unit is amplified, forming a concatemer through rolling circle replication, which exists in this form only transiently. (3) High frequency recombination at the region of shared context which was also suggested by *Bos et al., 1980* produces nearly uniformly random orientations of alignments in a concatemer form. This proposed mechanism, and the fact that seven colonies derived from the passaging of one spontaneous Petite all had this 'mixed' structure, strongly suggests that cells in these colonies are heteroplasmic in these various structures because they are not readily segregated. As such, this structure represents a unique example of coexisting structural isoforms in the mtDNA of baker's yeast, that are produced through rapid recombination events that counteract the periodic structures produced by rolling-circle replication and any strong selection for particular configurations of mtDNA.

## How is the observed structural heterogeneity of mtDNA in yeast colonies partitioned among individual cells?

Given the evidence of heteroplasmy in 'mixed' structure colonies, we were curious to understand the nature of the low frequency alternate mtDNA structures we observed in both Grande and Petite colonies. In the bulk sequencing of colonies, low frequency structures can be contributed by both heteroplasmic cells and mixed populations of cells homoplasmic for primary and alternate structures. In the heteroplasmic limit (*Figure 6a*), the majority of alternate structure content in a colony is contributed by cells in a heteroplasmic state. One consequence of being in this limit is that biological replicates of colonies would be expected to have low variance in total alternate structure content if heteroplasmy persists. In the homoplasmic limit, the major contribution to total alternate structure content comes from cells solely containing alternate structures (*Figure 6b*). In this limit, stochasticity in the time of generation of the mutational event would be expected to result in high variance in alternate structure content across biological replicates. Furthermore, in circumstances that enable differential selection on homoplasmic lineages, such as in Petite lineages within Grande colonies in non-fermentable conditions, one would expect the alternate structure content to change as a function of growth conditions if there were homoplasmic contributions.

In 10 Grande colonies sequenced, 4 were grown in YPG (non-fermentable media) and 6 in YPD (fermentable media). Only one colony in each growth condition harbored high confidence Petite concatemer structures according to our structural detection pipeline, represented by the two distinct breakpoint clusters in the lower half of *Figure 2d*. One of these high confidence structures within a YPG colony is shown in *Appendix 1—figure 5*. With a spontaneous Petite frequency of 10% in the genetic background of the strains sequenced (*Dimitrov et al., 2009*), these low detection rates are due to our conservative approach to detecting breakpoints. In our pipeline, we require at least three breakpoints from three separate reads to form a believable cluster of breakpoint signals (see Materials and methods). In Grande colonies that produce a diverse set of Petite structures afforded by excisions of the intact WT genome, forming high confidence breakpoint clusters, let alone clusters themselves, is unlikely. Therefore, to compute alternate structure frequencies in this analysis, we abandoned the requirement of breakpoint signals to form a cluster. Instead, we simply counted the total base-pair contribution of reads that included any detected breakpoints internal to the primary alignments as long as they were not accompanied by inverted duplication artifacts which are known to introduce

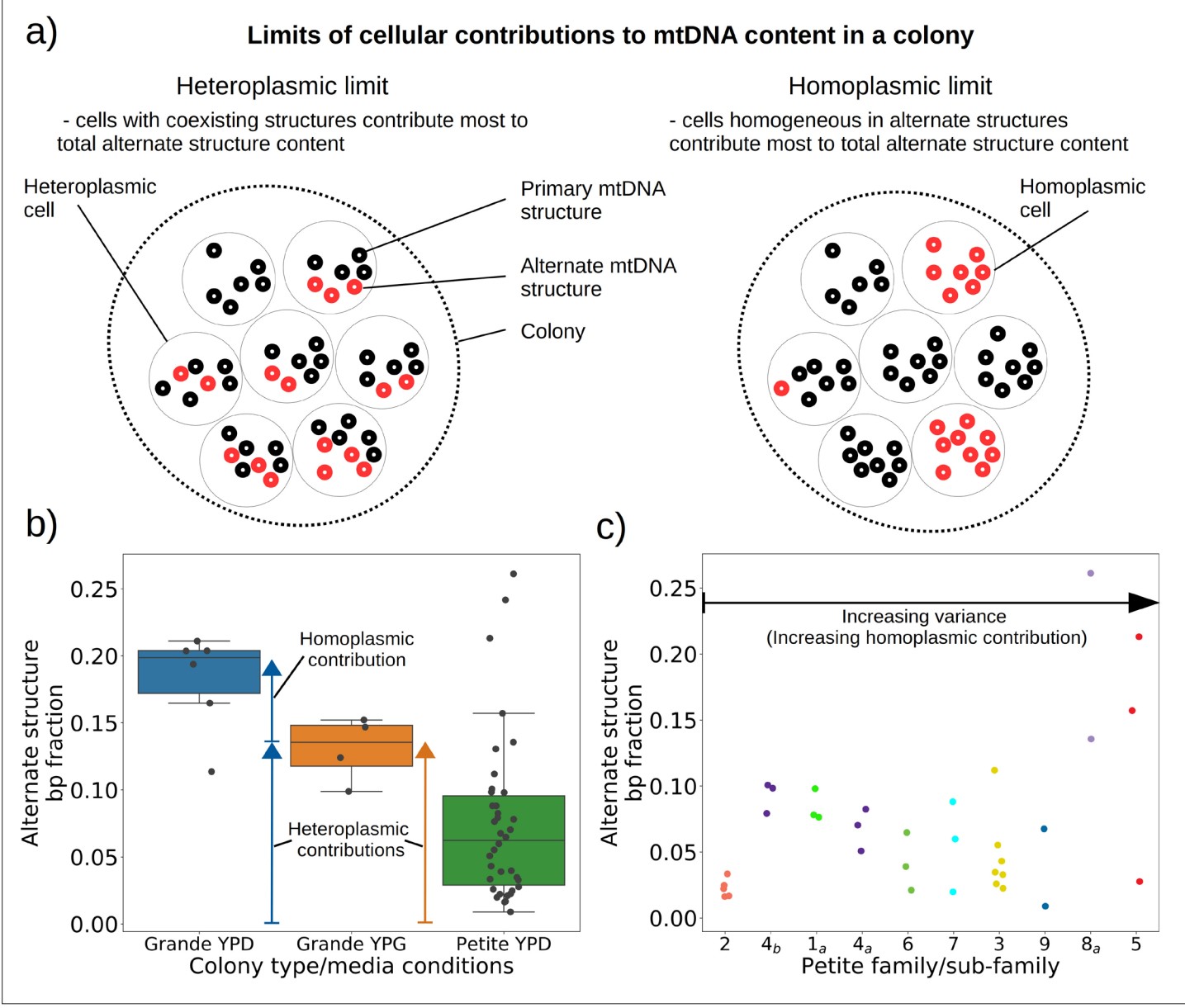

**Figure 6.** Evidence of heteroplasmy and homoplasmy in Grande and Petite colonies. (**a**) A schematic of the two limits of cellular contributions to mtDNA content in a colony. Left: In the heteroplasmic limit, most of the contribution to total alternate structure content comes from cells containing coexisting alternate and primary structures. Right: In the homoplasmic limit, most alternate structure content comes from cells homogeneous in alternate structure content. (**b**) The total fraction in bp of reads that include any breakpoint not expected from the primary structure in Grande samples in YPD (fermentable carbon source), YPG (non-fermentable), and Petite samples in YPD. Each dot represents the alternate structure content fraction for a single colony, which is the fractional contribution to total mitochondrial content of reads that contain breakpoints that differ from breakpoints in the primary structure. The box plot displays the median value, and the minimum, maximum, first quartile, and third quartile. Blue vectors indicate heteroplasmic/homoplasmic contributions in Grande colonies in YPD. The orange vector indicates heteroplasmic contributions in Grande colonies in YPG. (**c**) Contributions of petite families/subfamilies to the Petite YPD alternate structure bp fractions in (**b**) sorted by variance in alternate structure basepair fractions in each subfamily. The arrow indicates that increasing variance is expected to be accompanied by increasing homoplasmic contributions.

spurious breakpoints due to noise in the latter half of the read (*Spealman et al., 2020*; *Appendix 1—figure 5*).

The results of this read enumeration approach (*Figure 6b*) provide two lines of evidence for heteroplasmy and homoplasmy in Grande cells when comparing mtDNA colony composition under fermentable/non-fermentable media. In Grande colonies grown under non-fermentable conditions (YPG), we argue that the presence of Petite (fragmented) mtDNA in bulk sequencing is contributed

by heteroplasmic cells or recent heteroplasmy (orange vector in *Figure 6b*). This is because in contrast to the heteroplasmic cells containing Grande mtDNA, cells homoplasmic for Petite mtDNA cannot replicate under non-fermentable conditions. With this notion, any observed Petite mtDNA in YPG resides either in cells with Grande mtDNA in a heteroplasmic state or is stuck in homoplasmic Petite cells which are unable to replicate but are a product of recent heteroplasmy. The second argument is that the observed increase in Petite mtDNA fraction that accompanies a switch to fermentable media (YPD) in *Figure 6b* is predominantly due to homoplasmic cells. Again, since under non-fermentable media homoplasmic Petites are suppressed, any increase in Petite mtDNA once we relax respiration requirements should primarily be due to homoplasmic Petite cells. Thus, these results in Grande colonies suggest that both heteroplasmic and homoplasmic cells (sum of blue vectors in *Figure 6b*) are contributing to alternate structures in fermentable conditions.

In Petite colonies where all cells regardless of mtDNA content are equally fit in fermentable media, we point to two lines of evidence indicating heteroplasmic and homoplasmic contributions to alternate structure frequencies in *Figure 6*. The following arguments are based on the median value of the alternate structure frequency that is shown in *Figure 6b*, and the variance of the same frequencies grouped into Petite families/subfamilies in *Figure 6c*. The first argument we make for evidence of heteroplasmy in Petite colonies is based on the observation that the median alternate structure frequency in Petites is lower than that of Grandes in *Figure 6b*. We suggest this is due to stronger out-competition of alternate structures by primary structures in Petites than in Grandes. Because small Petite primary structures replicate much faster than Grandes, alternate structures in Petites are less likely to take hold in this competition. Under this assumption, part of the alternate structure signals must be due to cells in at least transiently heteroplasmic states to enable this competition. Using *Figure 6c* and known properties of mtDNA transmission, we also argue that most variance in alternate structure frequencies within Petite families is due to stochasticity in the generation of homoplasmic lineages. This is because mtDNA transmission bottlenecks and within-cell selection favor the production of homoplasmic clones containing alternate structures (*Ling and Shibata, 2004*). However, we

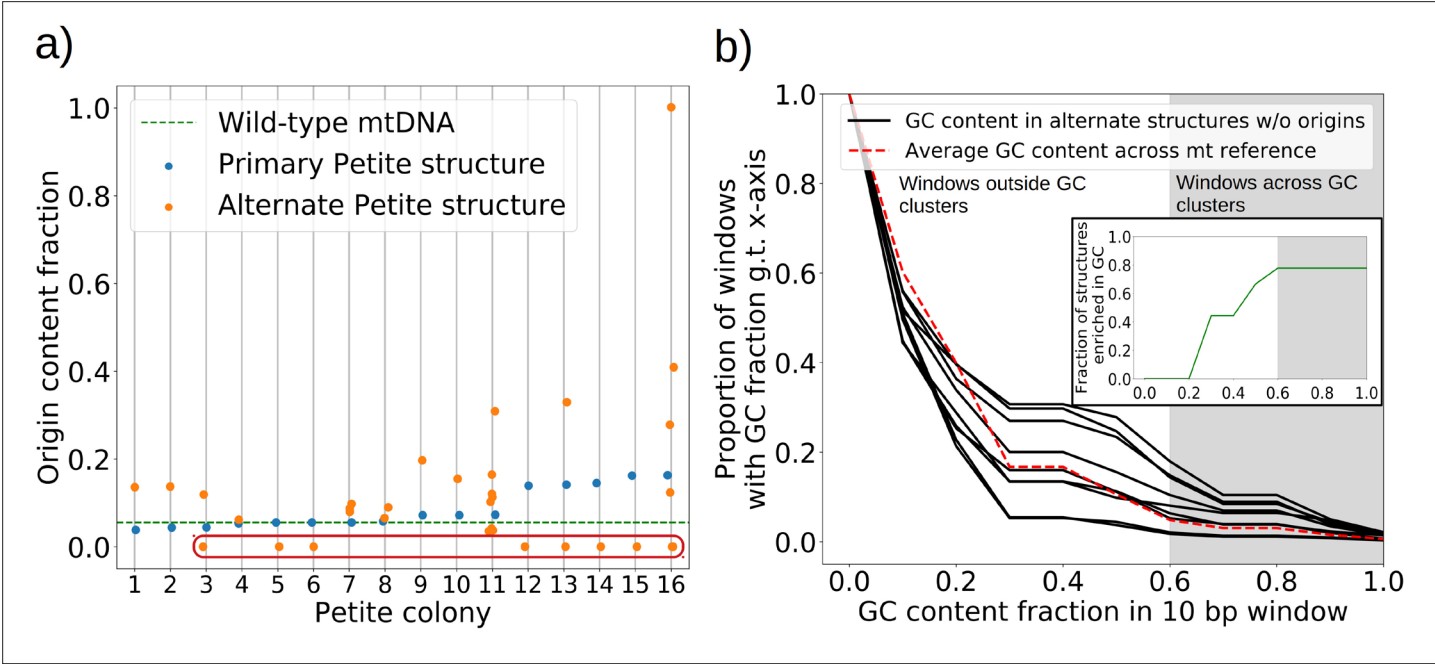

**Figure 7.** The role of replication origins and GC clusters in mtDNA replication. (**a**) Replication origin content fractions in primary/alternate structures detected in all samples where both are present. Each dot represents the base pair fraction of any of the eight origins of replication in detected structures. Orange dots are the origin fractions in alternate structures, blue in primary structures, and the green line is the origin content fraction in the wild-type mitochondrial genome. Highlighted by a red bubble are nine alternate structures that are devoid of an origin of replication. (**b**) Black curves (nine total) represent the cumulative distribution of GC content fraction in a sliding window of 10 bp in the highlighted zero-ori alternate samples. The red curve highlights this same GC distribution but in the wild-type (Grande) mitochondrial reference. The gray region indicates GC content fractions in sliding windows that are consistent with GC clusters found in replication origins (*Appendix 1—figure 8*). The inset shows the fraction of black lines above the red line as a function of GC fraction in the 10 bp window.

also highlight that some families have distinguishably lower variance in alternate structure fractions in *Figure 6c* (e.g., 2 and 4$_b$), which is at odds with this hypothesis. As such, these low variance alternate structure fractions in select Petite families appear to indicate persistent heteroplasmic contributions. Thus, as in Grande colonies, we are able to tease apart indications of heteroplasmic and homoplasmic contributions to mtDNA diversity in Petite colonies.

## What contributes to the fitness of mtDNA structures, and how does structure inform suppressivity?

### The role of origins of replication and GC clusters in mtDNA replication

The mechanism of generation of Petite mtDNAs, as well as our explanation of the distribution of excisions observed in spontaneous Petites, relied on the presence of replication origins. Given their importance in conferring a replication advantage to Petite mtDNAs, we were interested to look for mtDNAs without replication origins that we expected would exist at a low frequency. We were also curious to know if mtDNAs devoid of replication origins had any shared structural characteristics that might explain their propagation. Consistent with the notion that repeated structures with high densities of replication origins have a selective replication advantage over wild-type Grande mtDNA (*de Zamaroczy et al., 1981*; *Bernardi, 2005*), 32 of the 47 repeat structures detected across the 16 Petite colonies that contain alternate structures exhibit a higher replication origin content fraction than wild-type mtDNA (*Figure 7a*). Furthermore, all primary structures contain at least a portion of an origin. However, some of the detected alternate structures encircled in red in *Figure 7a* have no replication origin content at all. These resemble the 'surrogate' replication origin structures described in *Goursot et al., 1982* and appear to contain GC clusters in similar configurations to replication origins (peaks above 0.6 GC content in *Appendix 1—figure 8*), which are known to be important for replication and transcription initiation (*Baldacci and Bernardi, 1982*; *de Zamaroczy and Bernardi, 1986*). Consistent with this idea, seven of the nine structures without replication origins are enriched in GC clusters compared to the average GC cluster content of wild-type mtDNA (*Figure 7b*). Besides the suggested involvement of GC clusters in replication, the enrichment in GC clusters here is also consistent with the observation that GC clusters themselves may be preferred over AT-rich regions as excision sites (*Faugeron-Fonty et al., 1979*; *Gaillard et al., 1980*). While structures without replication origins are rare in cultured spontaneous Petites (*Goursot et al., 1982*), high depth long-read sequencing has provided access to these low frequency structures. The ease of identification of these mtDNA structures through long-read sequencing and accompanying structural inference techniques may prove useful in exploring the minimal sequences required for replication in yeast, as well as low frequency genome diversity in other systems.

### How mtDNA structure informs suppressivity

To understand the rules of competition between wild-type and mutant mtDNA, we measured the suppressivity of all Petite colonies within families (see Materials and methods). Suppressivity is a measure of the fraction of Petite progeny in a cross between each Petite sample and a Grande tester strain. Unlike previous work that studied the relationship between structure and suppressivity in highly suppressive Petites with suppressivity upwards of 90% (*de Zamaroczy et al., 1981*), our strains exhibit suppressivities from the basal rate of the Petite frequency of the Grande strain at 10%, to ~90%, and these suppressivities correlate well with repeat unit lengths of up to 70 kbp (*Appendix 1—figure 9*). In contrast, the repeat units in previous work were smaller than 10 kbp. While this difference in repeat unit size was due to the intentional selection of small repeat units in the previous work, distributions of deletion sizes and therefore observed suppressivities have been shown to be dependent on numerous nuclear genes (*Bradshaw et al., 2017*; *Ling et al., 2019*). To describe how the structure of mtDNA in our samples informs suppressivity, we developed a phenomenological model (*Figure 8a*) which assumes each repeat unit is independently competing (we discuss alternate models in *Appendix 1—figure 10*). The key assumptions of the model that explains the data well are: (a) in mating both Grande and Petite cells contribute equal mitochondrial content, $M$, which is motivated by the observation of equal Grande and Petite contributions observed in *MacAlpine et al., 2001*, (b) the number of repeat units initially contributed by Grande and Petite cells during mating is given by $M$ over the average repeat unit length ($L_G$ or $L_P$, G for Grande, $P$ for Petite), and (c) Petite and Grande repeat units replicate independently and exponentially with a replication rate linearly dependent on a

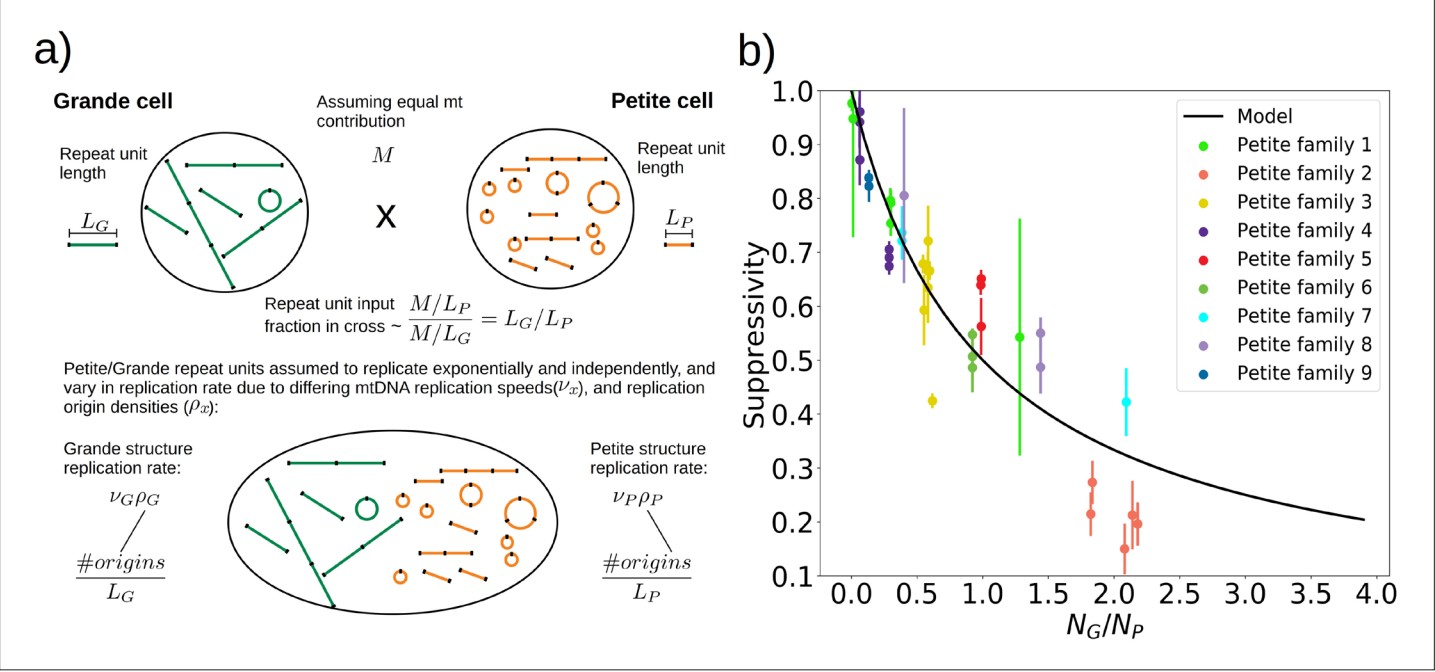

**Figure 8.** A phenomenological model of suppressivity. (**a**) A visual depiction of a phenomenological model of suppressivity. Grande and Petite cells are assumed to contribute equal quantities of mtDNA. It is also assumed that each repeat unit replicates independently and exponentially and that during mating the repeat unit input fractions of Grandes and Petites are inversely proportional to repeat unit length. Exponential growth rates are the product of mtDNA replication speeds and origin densities. (**b**) Suppressivity of all samples compared to a fit of *equation (1)*, which is the black line. The fit parameters are $\nu_g t^* = 10677$ bp, $\nu_p t^* = 2296$ bp, and the coefficient of determination is $R^2 = 0.85$. Dots are the average suppressivity across three second passage Petite colonies that share the same first passage progenitor and belong to the families indicated in the legend (same colored dots share a spontaneous Petite colony progenitor). Y-axis error bars are ± the standard deviation in suppressivities across these three second passage Petites colonies. Samples containing inverted breakpoints in their primary structure are those derived from families 2 and 3, the orange and yellow dots, respectively. Family 3 is the mixed structures described in the text.

mtDNA replication speed ($\nu_G$, $\nu_P$) and replication origin density ($\rho_G$, $\rho_P$). The suppressivity is then the fraction of Petite repeat units after a certain competition time (t) and is given by the ratio of the time evolution of an exponentially growing population of Petite repeat units $N_P = (M/L_P) e^{\nu_P \rho_P t}$ to total repeat units $N_P + N_G = (M/L_P) e^{\nu_P \rho_P t} + (M/L_G) e^{\nu_G \rho_G t}$ given in *equation (1)*:

$$\text{Suppressivity} = \frac{1}{1 + \frac{N_G}{N_P}} = \frac{1}{1 + \frac{L_P}{L_G} e^{(\nu_G \rho_G - \nu_P \rho_P)t}} \tag{1}$$

The data and least-squares fit of this model, with $\nu_G t^*$ and $\nu_P t^*$ as fit parameters, are shown in *Figure 8b*. These fit parameters are the products of mtDNA replication speeds in Grandes and Petites with the competition time ($t^*$) over which competition of Grande/Petite structures occurs following mating. The repeat unit lengths $L_G$ and $L_P$ in *equation (1)* are taken to be the sum of unique alignment lengths in each sample, and $N_G/N_P$ is the second term in the denominator of *equation (1)*, which is the ratio of the Grande to Petite fragment population. To understand the values of the fit parameters and whether or not they are reasonable, we compare them to equivalent parameters inferred from an exponential growth model of a budding Grande cell: First we assume that the mtDNA competition window ($t^*$) is equal to the doubling time in a diploid Grande population of cells (90 min). This is a reasonable assumption, as zygotes generally give rise to their first bud within 90 min in our mating experiments and early zygote dynamics dominate suppressivity results. If we also assume that the exponential replication rate of mtDNA in the Grande cell is the product of replication origin density (1 every 10 kbp in Grandes) and replication speed, then the average mtDNA replication speed in Grande cells is 82 bp/min if mtDNA is duplicated over the cell doubling time. This is of the same order as $\nu_G$ and is within an order of magnitude of $\nu_P$ in *Figure 8b*. We note, however, that these mtDNA replication speeds are coarse grained parameters in the model and should not be compared to directly

measured DNA fork velocities which require careful consideration of numerous biological parameters (**Hawkins et al., 2013**).

With respect to the architecture of the model, a variety of alternative models were also tested in **Appendix 1—figure 10**, revealing that both exponential growth and a repeat unit input fraction inversely proportional to repeat unit length are statistically important inclusions in improving the model in most regimes. The inverse repeat unit length terms seem to suggest that within yeast zygotes, early competition operates in a repeat unit limit where concatemers are reduced to monomeric forms which then undergo replication. Interestingly, active concatemer to monomer partitioning has been observed during mitosis in yeast (**Ling and Shibata, 2002**; **Ling and Shibata, 2004**; **Ling et al., 2007**), although to our knowledge little is known about the structure of mtDNA during mating and zygote formation. Thus, according to this model, the rules of competition between wild-type and mutant mtDNA in yeast depend on the exponential replication of monomeric forms of mtDNA in zygotes, where replication rates are proportional to replication origin densities in repeat units. This highlights the possibility that Petite mtDNAs may have both a replication advantage and segregational advantage if replication occurs in physically separated repeat units in zygotes.

## Discussion

In this article, we studied the dynamics of mtDNA fragmentation in yeast through long-read sequencing and quantified Petite mtDNA fitness through mating experiments. The use of long-read sequencing technology, in conjunction with structural inference methods we developed, which to our knowledge have never been applied to Petite mtDNA in yeast, gave us the ability to reconstruct complex mtDNA structures within populations of growing Petite colonies. This experimental approach enabled us to answer some important open questions about Petite mtDNA formation and propagation. On Petite mtDNA generation, we discovered contingency as a driving force behind mtDNA excision dynamics where previous fragmentation sites seed new events. This along with evidence for the generation of non-periodic 'mixed' mtDNA structures shows the power of our approach to understand structural variants and their dynamics. On Petite mtDNA propagation, this article reinforced that within cell (intracellular) selection plays a key role in the fragmentation dynamics of Petite mtDNA. A replicative advantage for mtDNA fragments with a high density of replication origins explains why mtDNA excisions tend to cluster near origins and was a critical component of the biophysical model of mtDNA fitness we developed. Both intracellular and cell-level (intercellular) selection also helped explain the distribution of altered mtDNAs among cells in a colony.

Building upon previous work, which alluded to ongoing mtDNA fragmentation in Petites, we provided direct evidence for this fragmentation in Petite colonies and discovered that subsequent mtDNA fragmentation is contingent on previous fragmentation. The presence of various levels of heterogeneity observed within Petite strains indicated by non-primary sub-stoichiometric bands in the restriction digests of Petite mtDNA (**Bernardi et al., 1976**; **Lewin et al., 1978**; **Lewin et al., 1979**; **Locker et al., 1979**; **Marotta et al., 1982**) suggested that the excision mechanism was ongoing, continuously producing lower complexity Petite structures in a hypothesized excision 'cascade' (**Locker et al., 1979**; **Bernardi, 2005**). Here, we have demonstrated unequivocal evidence of secondary excisions that operate on primary structures in sequenced colonies, and unlike the previous work, were quantitative in computing the frequencies of these coexisting structures. We also showed, for the first time, that subsequent excisions are contingent on previous excisions that produced the primary structure in these colonies. This apparent preference for the reuse of existing excision locations constrained the fate of structures formed through subsequent excisions. The reuse of excision sites highlights a tension between contingency and repeatability in the formation of new Petite mtDNA structures. It also seems to suggest that the breakpoints in primary Petite structures are persistent instabilities in mtDNA, perhaps akin to structures like R-loops (**Holt, 2019**) that may promote strand invasion and recombination at or near these sites. Exploration of the nature of these instabilities in the mtDNA of yeast remains an interesting direction for future studies.

Then we established that previously hypothesized non-periodic 'mixed' mtDNA structures are real and indeed non-periodic, which is at odds with the structures of most Petite mtDNAs observed. Hints of non-periodic or non-tandemly duplicated structures ('mixed' structures) have been commented on previously in ethidium bromide treated Petites (**Locker et al., 1974**; **Locker et al., 1979**) and spontaneous Petites (**Heyting et al., 1979**; **Bos et al., 1980**; **Faugeron-Fonty et al., 1983**). The first

proposal of a model for the generation of these structures was provided in *Bos et al., 1980*, but was subsequently refuted in *Faugeron-Fonty et al., 1983*,- where the claim was that these were larger ranging periodic structures produced by an unknown mechanism. However, our evidence of partially repeated units in *Figure 5d*, where two alignments are repeated with the same orientation but separated by an inverted alignment, precludes the structure proposed in *Faugeron-Fonty et al., 1983*. Thus, this structure is indicative of rapid recombination within mtDNA concatemers that opposes the homogeneity produced by rolling-circle replication and segregation through bud bottlenecks, which results in a collection of coexisting structural isoforms of mtDNA. While coexisting concatemers of various lengths and forms have been observed in yeast mtDNA with the same repeat units (*Locker et al., 1979*), as well as coexisting isoforms in plant mitochondria (*Kozik et al., 2019*), the 'mixed' structures we have observed are a rare glimpse of this phenomenon in yeast that provide a unique example of persistent intramolecular and intermolecular heterogeneity. These structures are also interesting from the perspective of reverse excision events thought to partition concatemers into monomers during bud formation (*Ling and Shibata, 2002*). It is unclear how a monomer should be defined in these 'mixed' samples, and therefore how it is partitioned, given that circular repeat units are the predominant species in new buds.

Our quantitative analysis, despite using bulk sequencing, allowed us to address an important unaddressed question of how mutant mtDNAs were distributed among cells within colonies. Persistent heterogeneity in Petite colony mtDNA was observed previously as sub-molar restriction digestion fragments (*Lewin et al., 1979*). These fragments were persistent across biological replicates but seen to disappear and reappear with varying intensities during subcloning. While their persistence indicated heteroplasmy, the varying intensities pointed toward clonal divergence through segregation into homoplasmic clones. Similarly, we argued that most of the variation in mtDNA structure we observed within colonies is likely due to homoplasmic clones in Petites, but with hints of heteroplasmic contributions in a few examples. Our observation of alternate structures in Grande samples under non-fermentable growth conditions is a direct indicator of heteroplasmy which is usually difficult to resolve from coexisting homoplasmic clones in bulk sequencing data. Recently, a variety of single-cell sequencing approaches have been adapted for use in yeast (*Jariani et al., 2020*; *Urbonaite et al., 2021*; *Dohn et al., 2021*) that are poised to enable direct observations of mtDNA heteroplasmy in yeast cells like they have in humans (*Maeda et al., 2020*). These tools will also provide an opportunity for quantification of mtDNA heteroplasmy (*Lareau et al., 2021*) which remains a promising direction for future work.

The inferred fine structures of Petite mtDNA from long reads allowed us to develop a phenomenological model for how structure informs fitness measured through the suppressivity of Petite samples. The relationship between suppressivity and mtDNA structure was explored in *de Zamaroczy et al., 1981*, which provided two general rules: (1) Partial deletions or rearrangements of origins of replication, including inversions of fragments containing origins, reduce suppressivity, and (2), suppressivity is inversely proportional to repeat unit length. This was followed with an observed exception to the second rule (*Rayko et al., 1988*), which suggested that flanking regions also influenced suppressivity.

In agreement with these rules, we showed that intact replication origins are indeed enriched in both primary and alternate Petite structures compared to wild-type, and that surrogate origins are present in rare alternate structures devoid of canonical origins. We also showed that selection drives mtDNA excision events in Petites to cluster near replication origins. The colocalization of mtDNA excisions and replication origins we observed is consistent with a recent study using short-read sequencing (*Osman et al., 2015*). The study of *Osman et al., 2015* suggested that the colocalization of origins and structural mtDNA variations may be due to replication origins themselves being recombination hotspots, resulting in preferential excisions at these locations. However, we demonstrated through comparisons to excision models that selection for small origin-containing fragments following excisions throughout the entire genome can explain the empirical excision distribution. At the same time, although we cannot see smaller mtDNA variations like SNPs in the present study, rare single-base changes at inferred excision sites in Petite strains have been observed close to origins in *de Zamaroczy et al., 1983* and in *Osman et al., 2015*. This may mean that like the larger structural variations we observe in Petites, small variations may also be observed to be clustered near replication origins just by virtue of these regions being strongly selected for, rather than preferential mutations at these locations. It is also important to note that the mtDNA recombination landscape in wild-type yeast

(*Fritsch et al., 2014*) also varies significantly from the excision distribution we observed. However, we attribute these differences to the selection of origin containing fragments in Petites, as well as differences between homologous recombination (between DNA molecules) and sequence-specific illegitimate recombination (within individual DNA molecules) responsible for excisions.

Consistent with the role of repeat unit length in defining Petite mtDNA fitness, the most predictive model of suppressivity was in limit which assumes that both Grande and Petite mtDNAs are independently replicating in their monomeric repeated units. We note, however, that the observed size distribution of concatemers between Grandes and Petites has been highly variable. Haploid Petite cells have been found to have both larger and smaller average molecular sizes than Grandes depending on the strain, and in both cases, harbor a pool of concatemers of various sizes (*Locker et al., 1979*). This seems to suggest that in haploid cells, as opposed to zygotes which we consider in the model, mtDNA competition may not be operating in the repeat unit limit. Nevertheless, if the numbers of independent mtDNA concatemers in haploid cells regardless of their size are inversely proportional to repeat unit length, this would be consistent with our model. In fact, in new buds, which all cells start as, concatemer to monomer partitioning has been observed in experiments (*Ling and Shibata, 2002*; *Ling and Shibata, 2004*; *Ling et al., 2007*). So even if during replication each monomer expands to different sized concatemers, the numbers of independent mtDNAs can still be inversely proportional to repeat unit length.

Given the emphasis in our suppressivity model on repeat unit size, but also notable outliers such as family 2 in *Figure 8b*, it is also possible that effective repeat unit sizes are dictated by the possible secondary structures for a given concatemer. Suppressivities in Petite family 2, which contain tandemly repeated inverted dimers, deviate most from the theoretical curve. Inverted sequences like those in family 2 would also be expected to form the hairpin structure hypothesized in the generative model of the mixed repeats. If this hairpin persists, it will consume directly repeated regions that are preferred in crossover events. The result would be a reduction in the density of repeated regions accessible to excision events, which upon eventual fragmentation would produce a larger effective repeat unit. This may in part explain the lower suppresivity of Petite family 2 from the theoretical value. Overall, more data is likely required to aggregate these apparent exceptions or outliers into a more encompassing model of suppressivity, but the present study provides a foundation of modern techniques and lessons to build upon in this goal.

Considering the diversity and destructiveness of the Petite mutations that this study revealed, it is also worthwhile to comment on the tolerance of the Petite mutation in yeast populations and why their mtDNA might have evolved to produce a structure susceptible to such destruction. Lab strains of *S. cerevisiae*, like the one investigated in this study, generally have higher Petite frequencies than feral yeast strains due to a collection of nuclear mutations (*Dimitrov et al., 2009*). However, feral yeast strains of *S. cerevisiae* still have highly repetitive mtDNA that is susceptible to excision, albeit at a lower frequency. A natural question is then, why did evolution yield such a structure? It has been suggested that the addition of repetitive origins and surrogate origin sequences may have conferred a replicative or transcriptional advantage to the wild-type genome (*de Zamaroczy and Bernardi, 1986*; *Bernardi, 2005*). It is also possible that high rates of recombination enabled by this repetition is advantageous for genetic complementation. Another possibility of having a highly recombinant DNA and machinery is for the destruction of invasive foreign DNA. Finally, population-level selection for respiring yeast cells also likely played a central role in opposing the negative effects of this mtDNA instability in populations, helping maintain intact mtDNA in Petite-positive yeast over evolutionary timescales. As such, mtDNA dynamics and the Petite mutation in *S. cerevisiae* is a wonderful example of how multilevel selection can shape the evolutionary trajectories of genomes.

Finally, we comment on the applicability of the findings in this study to other organisms. We motivated in this article that long-read sequencing and the structural inference methods we developed were able to reconstruct complex coexisting mtDNA structures in yeast colonies. This methodology will also be beneficial for the exploration of other systems that contain complex and repetitive mtDNA structures. A promising area of use is in plants, where complex mtDNA isoforms have been shown to coexist within cells (*Kozik et al., 2019*).

Interestingly, it turns out that the same components of the process that lead to mtDNA deletions in yeast—recombination followed by excision, selection, and persistent instability—also lead to mtDNA deletions in the human tissues. These mtDNA deletions in humans have been seen to accumulate

during aging in skeletal muscle and brain tissue (*Fayet et al., 2002*; *Kraytsberg et al., 2006*; *Payne and Chinnery, 2015*) and are also associated with a variety of diseases including Parkinson's (*Chan, 2006*; *Bender et al., 2006*). Like in yeast, excisions of mtDNA due to recombination between repeated homology have been suggested to be the cause of large mtDNA deletions in humans (*Guo et al., 2010*). Remarkably, the 'common deletion' in humans, which is a ~5 kbp deletion delimited by interacting 13 bp repeats, appears to be most frequent because of selection for replication origins. While numerous repeats exist in human mtDNA, and result in a multitude of deletions associated with disease, the 'common deletion' retains both replication origins unlike lower frequency deletions (*Samuels et al., 2004*). This suggests that the most common mutant mtDNA propagated in human cells is also governed by the same type of intracellular selection for replication origins that drives the Petite mutation. Persistent mtDNA instabilities in human mtDNA, which are suggested to be due to mtDNA content inducing replication fork stalling, have also been observed to create recombination hotspots and colocalize with mtDNA deletions (*Kraytsberg et al., 2004*; *Phillips et al., 2017*). This type of instability studied in humans is precisely the type of event that may help explain the contingency in mtDNA fragmentation we observe in yeast.

# Materials and methods

## Key resources table

| Reagent type (species) or resource | Designation | Source or reference | Identifiers | Additional information |
| --- | --- | --- | --- | --- |
| Strain, strain background (*Saccharomyces cerevisiae*) | W303 | GenBank: JRIU00000000.1 | *MATa/MATα leu*2-3,112 *trp*1-1 *can*1-100 *ura*3-1 *ade*2-1 *his*3-11,15 | |
| Strain, strain background (*S. cerevisiae*) | yCO362 | Boris Shraiman lab at UCSB /GenBank: JRIU00000000.1 | *MATa* W303 *leu*2-3,112 *can*1-100 *ura*3-1 *ade*2-1 *his*3-11,15 | |
| Strain, strain background (*S. cerevisiae*) | SY2081 | Grant Brown lab at UofT /GenBank: JRIU00000000.1 | W303 *MATα leu*2-3,112 *can*1-100 *ura*3-1 *ade*2-1 *his*3-11,15 *trp*1-1 | |
| Strain, strain background (*S. cerevisiae*) | 10T3 | This study | W303 *MATα leu*2-3,112 *can*1-100 *ade*2-1 *his*3-11,15 *trp*1-1 | |
| Commercial assay or kit | Qiagen 20/G Genomic-tip | QIAGEN | Cat. no./ID: 10223 | |
| Commercial assay or kit | MinION Mk1B with Starter Pack | Oxford Nanopore | Starter Pack (Flow Cell FLO-MIN106 R9.4.1) | |
| Commercial assay or kit | EXP-NBD104 and EXP_NBD114 Native barcoding expansion | Oxford Nanopore | EXP-NBD104 EXP_NBD114 | |
| Commercial assay or kit | SQK-LSK109 Ligation sequencing kit | Oxford Nanopore | SQK-LSK109 | |
| Commercial assay or kit | AMPureXP purification and cleanup kit | Beckman Coulter | A63881 | |
| Software, algorithm | Minimap2 | *Li, 2018* | Minimap2 | |

## Yeast strains and their construction for suppressivity testing

The Grande tester strain used in mating with Petites was the baker's yeast strain yCO362 W303 *MATa leu*2-3,112 *can*1-100 *ura*3-1 *ade*2-1 *his*3-11,15, which was a gift from the Boris Shraiman lab at UCSB. To construct the Grande progenitor of Petite strains, we restored URA3 function in SY2081 W303 *MATα leu*2-3,112 *can*1-100 *ura*3-1 *ade*2-1 *his*3-11,15 *trp*1-1 which was a gift from the Grant Brown lab at the University of Toronto. To this end, we grew an *Escherichia coli* strain harboring pFA6a-URA3, which was a gift from Jon Houseley & David Tollervey (Addgene #61924). Plasmids were extracted and the URA3 fragment PCR amplified with primers that share 20 nt of short flanking homology with the reference yeast mitochondrial genome following the standard short flanking homology targeted recombination method (*Petracek and Longtine, 2002*). Expected PCR fragment sizes were confirmed on a gel and then transfected into SY2081 using the high-efficiency LiOAC yeast transformation protocol (*Brown et al., 2015*). Integration at the expected location was confirmed through PCR of flanking regions overlapping each breakpoint, and Sanger sequencing. To ensure integration was exclusive to our target location, we then performed tetrad analysis on the transformed SY2081

(named 10T3 hereafter) × yCO362 and observed 2:2 segregation as expected for a single integration site.

## Media and growth conditions

Both Grande and Petite colonies were cultured in YPAD medium (1% yeast extract, 2% bacto-peptone, 2% glucose, and 0.072% adenine hemisulfate). Petite colonies were detected under growth in YPADG medium (1% yeast extract, 2% bacto-peptone, 0.1% glucose, 3% glycerol, and 0.072% adenine hemi-sulfate). With a reduced glucose content, Petite colonies appear smaller and more translucent than their Grande counterparts in this media and the differences were discernible beyond 4 days at 30°C (*Dimitrov et al., 2009*). YPG medium (1% yeast extract, 2% bacto-peptone, and 3% glycerol) was used in culturing a subset of Grande colonies and verifying the respiratory deficiency of identified Petites. To measure suppressivity, we used SC-ura-trp (DG carbon source) media (0.67% bacto yeast nitrogen base w/o amino acids, 0.1% glucose, 3% glycerol, 0.2% dropout powder lacking uracil and tryptophan) which selects for zygotes due to strain auxotrophies. Liquid cultures were grown at 30°C in a linear shaking water bath, while solid media growth took place in a forced air incubator also at 30°C.

## Isolation of spontaneous Petite colonies

Liquid culture of strain 10T3 inoculated in YPAD media was washed in $dH_2O$ and plated on YPADG media. Respiratory deficiency of these Petite colonies was confirmed through replica plating on to YPG, as well as patching onto separate YPG plates. Following confirmation of these colonies being Petites, nine different colonies were streaked onto YPAD agar as a first passage. Three colonies were randomly selected from this first passage plate and streaked again onto YPAD agar, constituting a second passage. Three colonies from each second passage plate were cultured, stored as frozen stocks, had their suppressivities measured, and a subset was sequenced.

## Yeast suppressivity assay

Cultures of Grande (yCO362) and Petite strains (10T3 derived) were grown overnight in YPAD liquid media. Cultures were diluted to 0.1OD and grown for 3 hr. Equal volumes of each culture were mixed and incubated at room temperature for 20 hr to allow for mating. A small aliquot of this mating mixture was observed under a hemocytometer to calculate appropriate dilutions for plating. The mating mixture was diluted and washed in $dH_2O$, then plated onto SC-ura-trp (DG carbon source). After 5 days at 30°C, Petri dishes were scanned, and the fraction of small to total colonies on these plates was recorded as the suppressivity of the strain, calculated based on an average of 250 colonies labeled per strain. The average standard deviation in suppressivity across strains related by the same first passage progenitor was ~5%.

## Nanopore sequencing

Three or more progeny from 9 separate spontaneous Petites derived from strain 10T3, and 10 Grande colonies under YPD (6) /YPG (4) liquid culture growth of the same strain were sequenced on an Oxford Nanopore MinION Mk1B (*Figure 1*). Whole genomic DNA from these 48 colonies following culturing was extracted using a modified enzymatic Hoffman-Winston DNA extraction protocol as described in *Boeke et al., 1985*. DNA was then purified with a Qiagen 20/G Genomic-tip and barcoded with the Oxford Nanopore EXP-NBD104 and EXP_NBD114 barcoding kits in conjunction with the SQK-LSK109 ligation sequencing kit with long fragment Agencourt AMPureXP purification, following manufacturer's instructions. Twenty-four barcoded samples were pooled at a time in two FLO-MIN106 flow cells with R9.4.1 chemistry. Sequencing generated a total of 8.5Gbases of reads across both flow cells within 24 hr, with a mean read length of ~6 kbp and maximum length of 120 kbp. About 429 Mbases of reads were mapped to mitochondrial DNA, resulting in an average coverage per primary structure of ~700 across all Petite samples sequenced.

## Read basecalling, alignment, and filtering

Raw Nanopore reads were basecalled and demultiplexed with the ONT Guppy package 3.1.5-1. Reads that passed default quality score filtering (>9 qscore) were aligned using Minimap2 with default parameters (*Li, 2018*) to the *S. cerevisiae* reference release R64-2-1_20150113 (*Engel et al., 2014*)

available from yeastgenome.org. Following initial alignment, unmapped regions in reads were recursively aligned to the reference sequence to combat the Z-drop heuristic, which exists to remove spurious alignment artifacts introduced due to the expectation of collinearity between alignment anchors. This is a particularly insidious heuristic when mapping Petite structures due to large numbers of repeats within reads. See *Appendix 1—figure 12*, where we provide an example of the effect of this Z-drop heuristic in resolving Petite structures, and how in a small subset of reads the pseudo-global alignment produced in our recursive mapping improves structure resolution. Alignments in reads were kept with PHRED scaled mapping quality scores>20, and alignment lengths>300 bp due to the high degree of homology between mt replication origins which makes alignment of these small regions of context difficult (*de Zamaroczy et al., 1981*).

## Mitochondrial DNA alignment breakpoint detection

Alignment breakpoints are defined in our pipeline to be deviations of more than 30 bp between the read and reference coordinates in adjacent alignments within one read. They are also identified by strand changes across adjacent alignments regardless of the separation in reference/read coordinates. As such, these breakpoint signals encompass large insertions, and deletions, and inversions within reads, and are delineated by alignment termination from the mapping program Minimap2 (with default parameters), which are stored and processed in a structural detection pipeline written in Python.

## Removal of prevalent inverted duplication artifacts

Before breakpoint signals can be clustered, inverted duplication artifacts, which are prevalent in our particular sequencing chemistry and are apparently affected by growth media (*Appendix 1—figure 1a*), must be filtered out from real inverted duplication breakpoint signals. These artifacts appear to be due to complementary strands being pulled in succession through the pore, resulting in an unfolding of a double-stranded DNA molecule into a molecule of double the length with a characteristic inversion. This may be due to either physical tethering or lingering of the separated complementary strand near the pore opening. In any case, conveniently this results in a nearly centered inversion within such reads (*Appendix 1—figure 1b*), that is sometimes skewed toward the end of the read because of increased translocation speed and self-interaction of the strands in a ratcheting mechanism as described in *Spealman et al., 2020*. This increased speed also results in reduced read quality in the latter half as reported in *Spealman et al., 2020* and *Appendix 1—figure 6a*, which means that distances between reference positions of alignments at inversions have the distribution in *Appendix 1—figure 6b*. Given that 90% of artifacts result in adjacent alignment edges with distances less than 1000 bp, we take this to be one of our criteria for an inversion artifact.

Inverted duplication artifacts are filtered from real inverted duplications at two levels before clustering: (1) Breakpoint signals where within read positioning of the inverted breakpoint is >1% likely to be derived from the purely artifact distribution in *Appendix 1—figure 1b* is recorded in a list. (2) If breakpoints that follow criteria (1) are derived from reads with only a single inverted duplication signal, then they are considered artifacts.

## Breakpoint clustering

Inverted/non-inverted breakpoints are clustered separately using the DBSCAN algorithm (*Ester et al., 1996*). This algorithm requires two parameters: $\epsilon$, which defines the neighborhood of a breakpoint as a radius in base pairs, and $\mathrm{minPts}$, which is the number points required within $\epsilon$ of a breakpoint to be considered a core point, or dense region. Breakpoints within dense regions are then connected together iteratively to produce a density-connected cluster. In case this density-based clustering inadvertently merges true clusters, k-means clustering, with *k*=2, is also performed within clusters output from DBSCAN. This handles the case where breakpoints transitions in a read occur within a DBSCAN cluster, indicating too coarse clustering. These marginal cases, however, only occurred in 3/35 alternate structures, and never in primary structures.

For both inverted and non-inverted breakpoints, we set $\mathrm{minPts} = 3$, to capture even the smallest clusters which will undergo further filtering described later. For non-inverted breakpoints, $\epsilon = 1$ kbp, which is a lenient choice given that the most catastrophic deviations in expected breakpoint positions due to sequencing error in inverted duplication artifacts are largely under 1 kbp. For inverted

breakpoints post artifact-filtering described above, an optimal $\epsilon$ is computed by sorting nearest-neighbor distances across all inverted breakpoints and computing the Euclidean distance to the nearest-neighbor that results in the largest curvature in a plot of nearest-neighbour distance versus breakpoint number. This is the so-called 'elbow' method. Then $\epsilon = \min\left(\text{optimal}\left(\epsilon\right), 1\text{kbp}\right)$ is taken to be the nearest-neighbor radius for inverted breakpoints. This extra step for inverted breakpoints is required solely because of the possibility of low frequency artifact noise that is not present for non-inverted breakpoints.

## Breakpoint filtering based on read support and a majority voting scheme

Following clustering with the above parameters, breakpoint clusters are then required to have a minimum of three separate reads supporting them to be considered real. Furthermore, in reads that contain a detected inverted duplication artifact, we use the duplicated signals on either side of this artifact to our advantage in a majority voting scheme: For breakpoints belonging to cluster $j$, we count the number of times a breakpoint belonging to $j$ is recapitulated on either side of an inverted duplication artifact, $P_j$, and the number of times it isn't, $N_j$. If $P_j > N_j$ we consider this breakpoint to be real. In the case that $P_j = N_j = 0$, meaning that the breakpoint does not exist in reads that contain inverted duplication artifacts, we perform a similar majority vote on the basis of a breakpoint being repeated or not within reads. The notion here is that real breakpoints should be repeated within reads because of the expected concatemer structure of Petites, while spurious breakpoints that are low frequency are less likely to be. To this end we compute $NR_j$ and $PR_j$, which are the number of times a breakpoint from cluster $j$ is present in a read with repeats but is not repeated itself, and the number of times a breakpoint assigned to cluster $j$ is repeated, respectively. Similar to the above, if $PR_j > NR_j$ we consider this cluster to be real.

See *Appendix 2—table 1* for a summary of all parameters in our pipeline described in this section and the three preceding it. See *Appendix 1—figure 13* for the effect that changing these parameters has on breakpoint counts. See *Appendix 1—figure 11* for the effect that this majority voting scheme has on removing spurious breakpoints.

## Breakpoint labeling and read encoding schemes

Breakpoints are labeled based on the segmentation provided by a Python implementation of DBSCAN (an integer), in addition to two other features: (1) the strand in this transition (±), and (2) the transition from a low (L) to high (H) or vice-versa reference base position. This means that read with a repeated excised repeat unit with the same orientation and in the +strand takes the form: [2LH, 2LH, 2LH, ..., 2LH] in the number transition encoding, and [2++, 2++, 2++, ..., 2++] in the orientation encoding scheme. In addition to these two schemes, the mapped ends of the reads in reference locations are also stored and used later in assigning reads to inferred structures. Both encoding schemes are necessary when considering complex structures such as the 'mixed' repeat structures described in results where only two breakpoints exist, but permutations in their orientation produce four unique alignments which would be missed with a simple numerical labeling scheme.

## Reconstructing mtDNA repeat structures from reads

For long reads containing small repeated structures with numerous breakpoints passing filtering criteria, repeats are detected by directly computing the longest common prefix within a read in both read encoding schemes described above. We require two periods to be present for this type of repeat detection method to produce a candidate structure, and that the detected repeat is a tandem repeat, meaning there are no intervening breakpoints between periods. This is how all low frequency alternate structures are detected within samples, and for some samples primary structures when small enough to be repeated within single reads.

Large primary structures that are too large to be fully repeated twice within reads are inferred through the construction of a breakpoint transition matrix, $T_{ij}$, that stores the number of transitions between breakpoint $i$ to breakpoint $j$ across all reads within a sample. If two entries in this matrix share a breakpoint, and if the fractional difference in their average counts to individual counts is <0.34, these breakpoint transitions are merged into a list together. This is a Binomial merging criterion for being within one standard deviation of the average counts assuming that both breakpoint transitions

sharing a breakpoint come from the same structure. Following recursive merging of transitions and then lists of transitions with the same criteria, if in each list the number of breakpoints equals the number of unique transitions (meaning this structure is a true repeat), then the structure is recorded alongside its average count and its span on the reference. Structures inferred from this method with the largest span on the reference are considered to be primary repeats, and if small enough are recapitulated by the direct repeat detection scheme described above.

## Mixed structure alignment frequency calculations

Consider an alignment of length $L_1$, where we are interested in computing its frequency in these mixed samples that contain four alignments with lengths $\{L_1, L_2, L_3, L_4\}$, accounting for read sampling bias. Given a read of length $x > L_1$, assuming sampling of a random pattern of all four alignments (or equivalently, random sampling of a periodic pattern including all four alignments), the probability this read is not truncating an alignment with length $L_1$ is $1 - \frac{L_1}{L_1+L_2+L_3+L_4}$ .

The estimated probability that we see this read of length $x > L_1$ is:

$$\int_{x=L_1}^{x=\infty} P_R(x)\, dx$$

where $P_R(x)$ is an exponential fit to the empirical read length distribution (*Appendix 1—figure 14*), with the following form, and is the read length probability distribution:

$$P_R(x) = \frac{1}{\beta} e^{\frac{-(x-\mu)}{\beta}}$$

where $\beta$ and μ are the scale and location parameters determined from the fit.

The joint probability that we have a read long enough to see an alignment, and that it does not abruptly truncate the alignment at its ends is the product of these two terms:

$$Q_{L_1} = \left(1 - \frac{L_1}{L_1+L_2+L_3+L_4}\right) e^{-\frac{(L_1-\mu)}{\beta}}$$

Alignment frequencies can then be computed by normalizing raw observed counts $N_i$ by $Q_{L_i}$, and computing their relative frequency, $\nu_i$ :

$$\nu_i = \frac{\frac{N_i}{Q_{L_i}}}{\sum_{i=1}^{i=4} \frac{N_i}{Q_{L_i}}}$$

## Primary/alternate structure frequency calculations

Unlike mixed samples where breakpoint identity does not uniquely define alignments (permutations of two breakpoints result in four alignments), in most samples we can simply count breakpoints, assuming that repeat units ~ unique breakpoint counts in a particular structure and perform a similar normalization to account for sampling bias.

To do this, each read is assigned to one class of potential structures based on the maximal overlap between breakpoint labels in known structures and the read, conditioned on containing content only within the known structure. Because we are counting breakpoints, we no longer have to worry about truncation of a whole alignment and can consider reads smaller than the expected alignment length.

Consider a repeat unit with period $L_k$ in bp, and a breakpoint count $J_k$ that is unique to this structure. The probability that we see this structure given the read distribution in this sample is:

$$Q_{L_k} = \int_{x=L_k}^{x=\infty} P_R(x)\, dx + \int_{x=u}^{x=L_k} \frac{x}{L_k} P_R(x)\, dx$$

The first term is the product of the probability (=1) that we see this breakpoint for read lengths $x > L_k$ and the probability that we have such reads ($P_R(x)$ is the read-length distribution fit as described above). The second term is the product of the probability we see this breakpoint for reads $x < L_k$ ($\frac{x}{L_k}$) and the probability that we see reads with these lengths.

The relative frequency can then be calculated by normalizing counts $J_k$ by $Q_{L_k}$ and computing relative frequency across all structures as in the previous section.

## Type I/II/III repeat unit classification

Type I repeat units are those excised from the interior of a primary Petite alignment. Type II repeat units contain alignments that span the primary alignment breakpoint when it is present in a concatemer form. Type III repeat units share one edge with the primary alignment and have another edge internal to the primary alignment. Because each edge in a reconstructed structure is truly a cluster of edge signals, for an alignment edge to be considered 'shared' the means of alignment edge location distributions must be equal to within one standard deviation of each other. Thus, according to this criteria, Type I repeats have no shared edges, Type II repeat units have two such shared edges and multiple alignments, and Type III repeat units have one shared edge and one alignment.

## Data availability

Raw Nanopore sequencing data (that has been demultiplexed and labeled with the corresponding colony name in the main text) is available alongside sequence alignment code and Python code for primary/alternate structure analysis. The data is available at https://doi.org/10.5061/dryad.vdncjsxwx. The code for analysis is available at https://doi.org/10.5281/zenodo.5851771. Preprocessed data and code to produce the plots in this article are available at https://github.com/javathejhut/ContingencyAndSelection, (copy archived at swh:1:rev:5b6f6c7e1fbff2a537f29d3b26bc292035170f6b; *Nunn, 2022*).

## Acknowledgements

The authors thank Dr. Grant Brown at the University of Toronto for his thorough reading of the manuscript and numerous members of the Goyal lab for their thoughtful comments and discussions. The authors also thank Dr. Boris Shraiman and Dr. Grant Brown for yeast strains. The authors received funding from the Natural Sciences and Engineering Research Council of Canada (NSERC) Discovery Grant RGPIN-2015-0, the Simons Foundation Grant 326844 in the Mathematical Modeling of Living Systems, and funding for equipment from the Canadian Foundation for Innovation (CFI) Grant 32708.

## Additional information

### Funding

| Funder | Grant reference number | Author |
| --- | --- | --- |
| Natural Sciences and Engineering Research Council of Canada | RGPIN-2015-0 | Sidhartha Goyal |
| Simons Foundation | 326844 | Sidhartha Goyal |
| Canada Foundation for Innovation | 32708 | Sidhartha Goyal |

The funders had no role in study design, data collection and interpretation, or the decision to submit the work for publication.

### Author contributions

Christopher J Nunn, Conceptualization, Formal analysis, Investigation, Methodology, Software, Validation, Visualization, Writing - original draft, Writing - review and editing; Sidhartha Goyal, Conceptualization, Funding acquisition, Investigation, Methodology, Project administration, Resources, Supervision, Writing - review and editing

### Author ORCIDs

Christopher J Nunn ⓘ http://orcid.org/0000-0002-9598-1045
Sidhartha Goyal ⓘ http://orcid.org/0000-0002-7452-892X

### Decision letter and Author response

Decision letter https://doi.org/10.7554/eLife.76557.sa1
Author response https://doi.org/10.7554/eLife.76557.sa2

## Additional files

### Supplementary files
• Transparent reporting form

### Data availability

Raw Nanopore sequencing data (that has been demultiplexed and labeled with the corresponding colony name in the main-text) is available alongside sequence alignment code and Python code for primary/alternate structure analysis. The data is available at https://doi.org/10.5061/dryad.vdncjsxwx. The code for analysis is available at https://doi.org/10.5281/zenodo.5851771. Preprocessed data and code to produce the plots in this article are available at https://github.com/javathejhut/ContingencyAndSelection, (copy archived at swh:1:rev:5b6f6c7e1fbff2a537f29d3b26bc292035170f6b).

The following dataset was generated:

| Author(s) | Year | Dataset title | Dataset URL | Database and Identifier |
|---|---|---|---|---|
| Nunn CJ | 2022 | Data and source code from: Contingency and selection in mitochondrial genome dynamics | https://dx.doi.org/10.5061/dryad.vdncjsxwx | Dryad Digital Repository, 10.5061/dryad.vdncjsxwx |

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

## Appendix 1

### Additional figures

a)

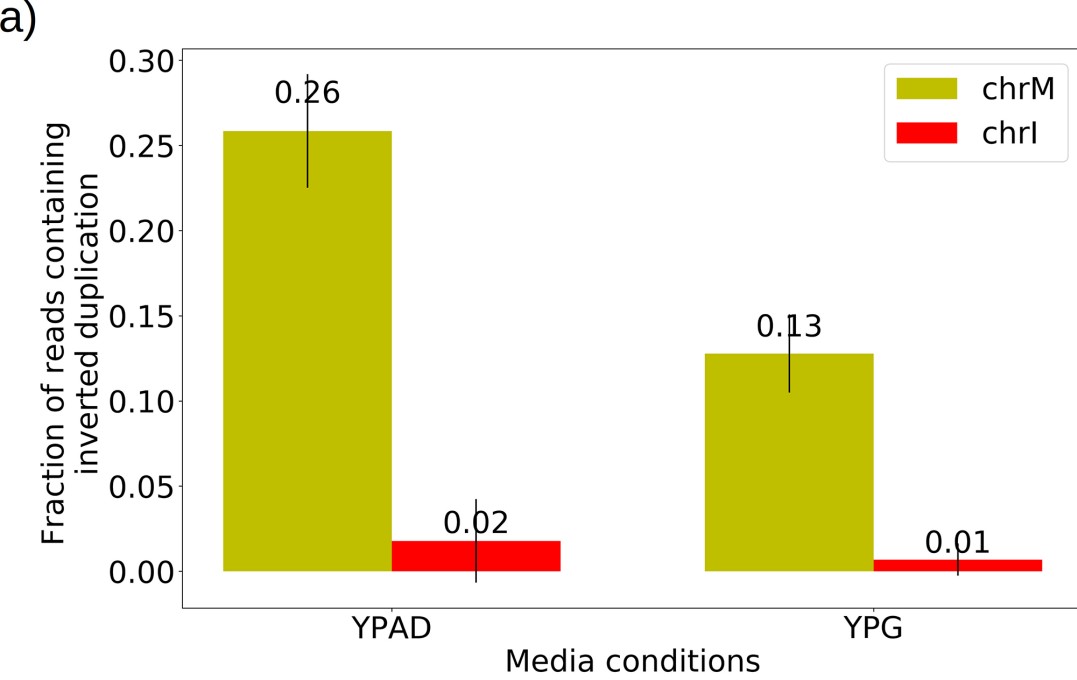

b)

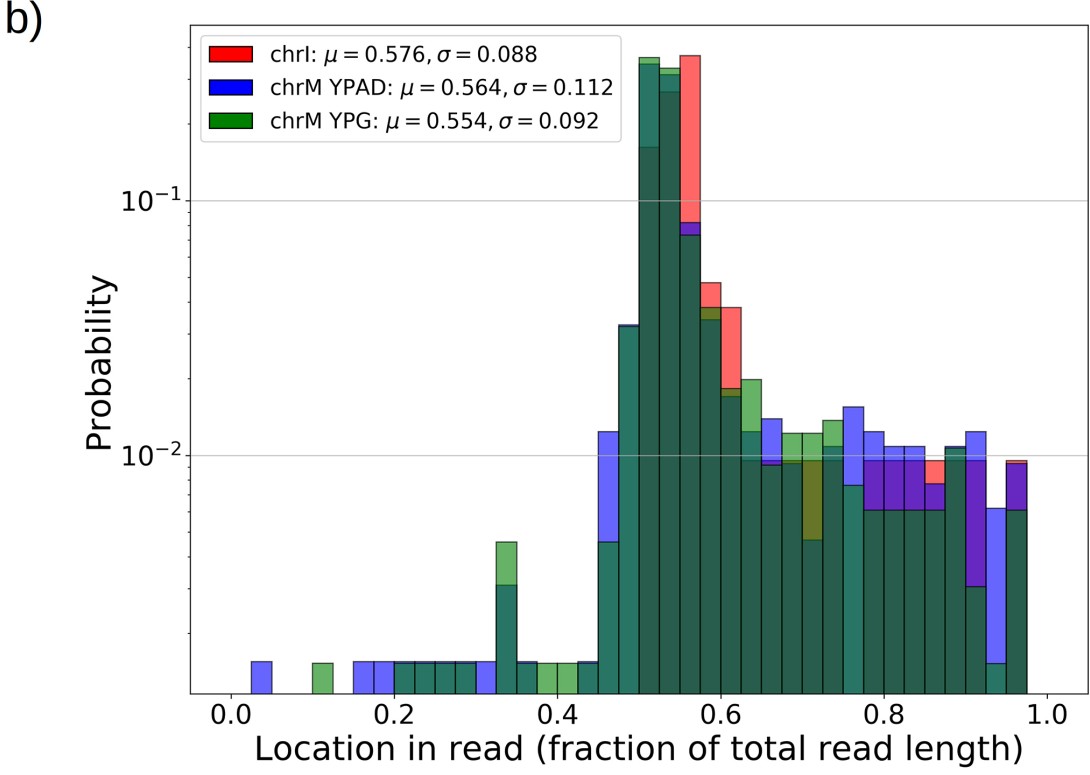

**Appendix 1—figure 1.** Inverted duplication artifacts are prevalent in Nanopore sequencing of mitochondrial DNA but exhibit patterns that enable their detection. (**a**) The fraction of reads containing repeated inverted annotated genome features were plotted above for all Grande colonies grown in YPD and YPG for both nuclear DNA (nDNA) and mtDNA. Inverted duplication artifacts were enriched in reads mapped to mitochondrial DNA. *Appendix 1—figure 1 continued on next page*

*Appendix 1—figure 1 continued*

While it has been suggested that these artifacts are caused by either tethered complementary strands being brought through the pore in series, or lingering complementary strands near the pore opening, it is unclear why there is such a difference between nDNA and mtDNA reads. Base composition does differ significantly between the two genomes, with only 18% GC content in mtDNA, compared to 38% in nDNA. The abundance of inverted duplication artifacts also appears to be affected by growth conditions, potentially due to differences in mtDNA conformation under respiration and fermentation conditions, but this remains unexplored. This effect was also seen in other Nanopore experiments performed by another lab with the same flow cell and sequencing chemistry described in Materials and methods (data not shown). (**b**) Here, we are plotting the probability distribution of the read locations of singleton inverted duplication artifacts in mtDNA across all Grande samples for both YPG and YPD conditions, as well as pooled reads from chromosome I (chrI) across both conditions. Mean and standard deviations of each set of samples are denoted in the legend. Artifact inverted duplications are generally concentrated toward the center of the read but biased slightly toward the second half. This is consistent with the self-interaction ratcheting mechanism described by *Spealman et al., 2020* in sequencing real inverted duplications, where self-interaction increases translocation speed in the second half of the read. Increased translocation speed results in skipped bases, effectively shortening the second half of the read which results in this bias to the right in fractional length. Inverted duplications detected are filtered to those residing within the 1% tails of this distribution.

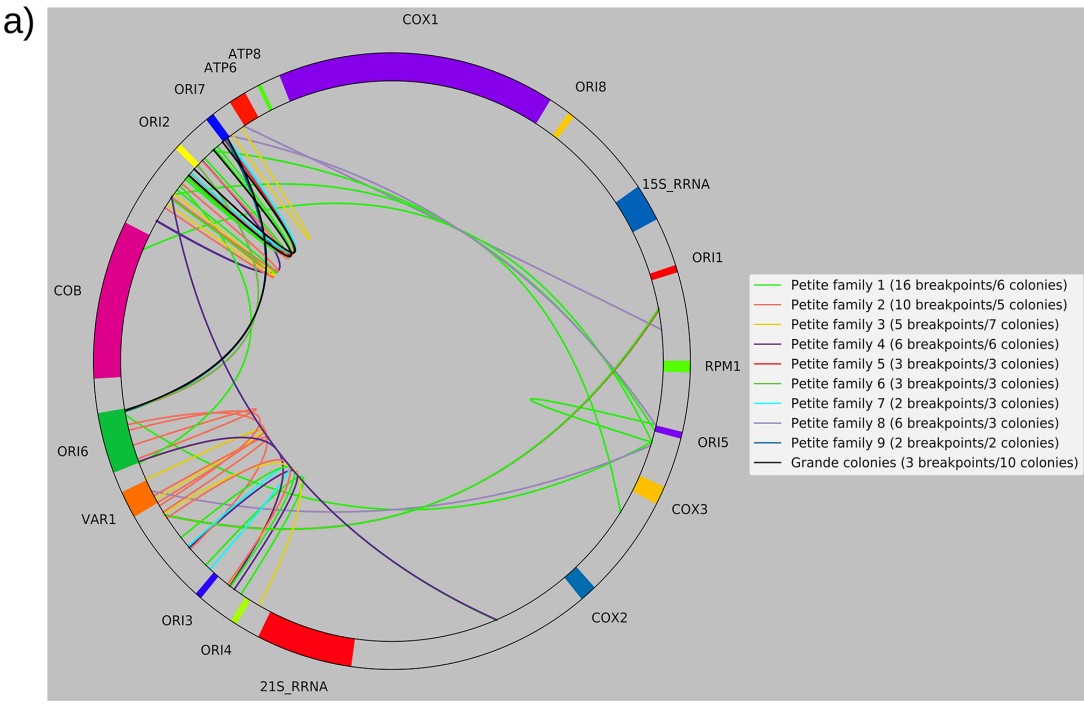

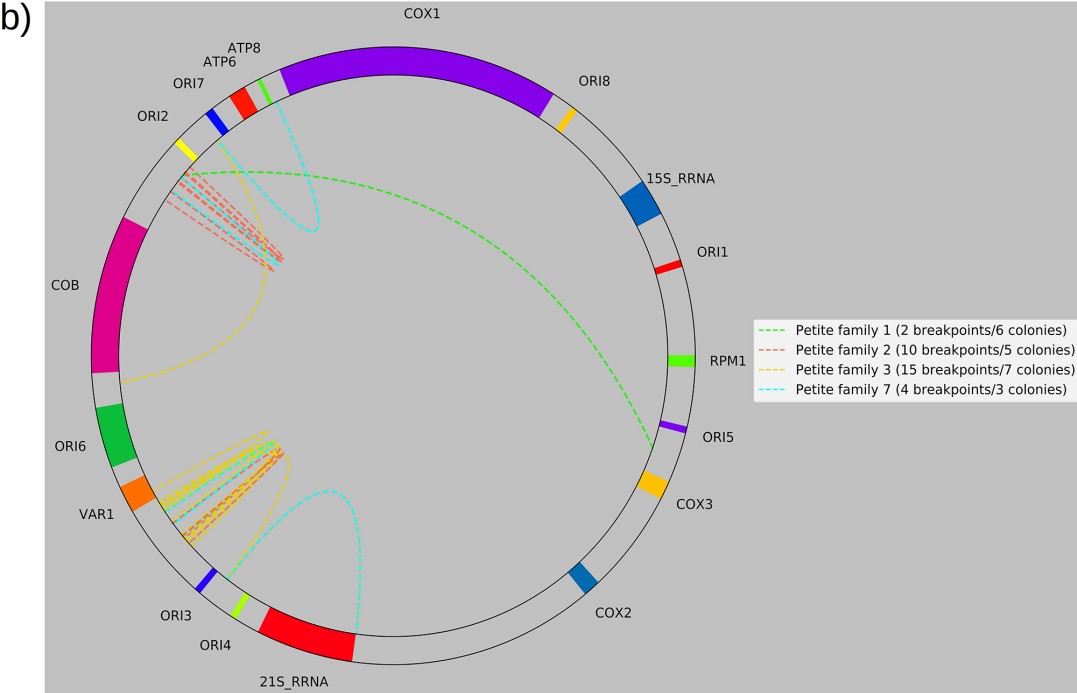

**Appendix 1—figure 2.** Separated non-inverted/inverted circular breakpoint plots. Here, we provide an alternate representation of the breakpoint summary in *Figure 2c* that separates non-inverted and inverted breakpoints and represents breakpoints as arcs to improve readability. The mitochondrial reference has also been circularized. (**a**) Non-inverted breakpoint locations. Here solid-colored arcs directly show the regions of mtDNA that interact in creating breakpoints. (**b**) Inverted breakpoint locations represented as dashed lines.

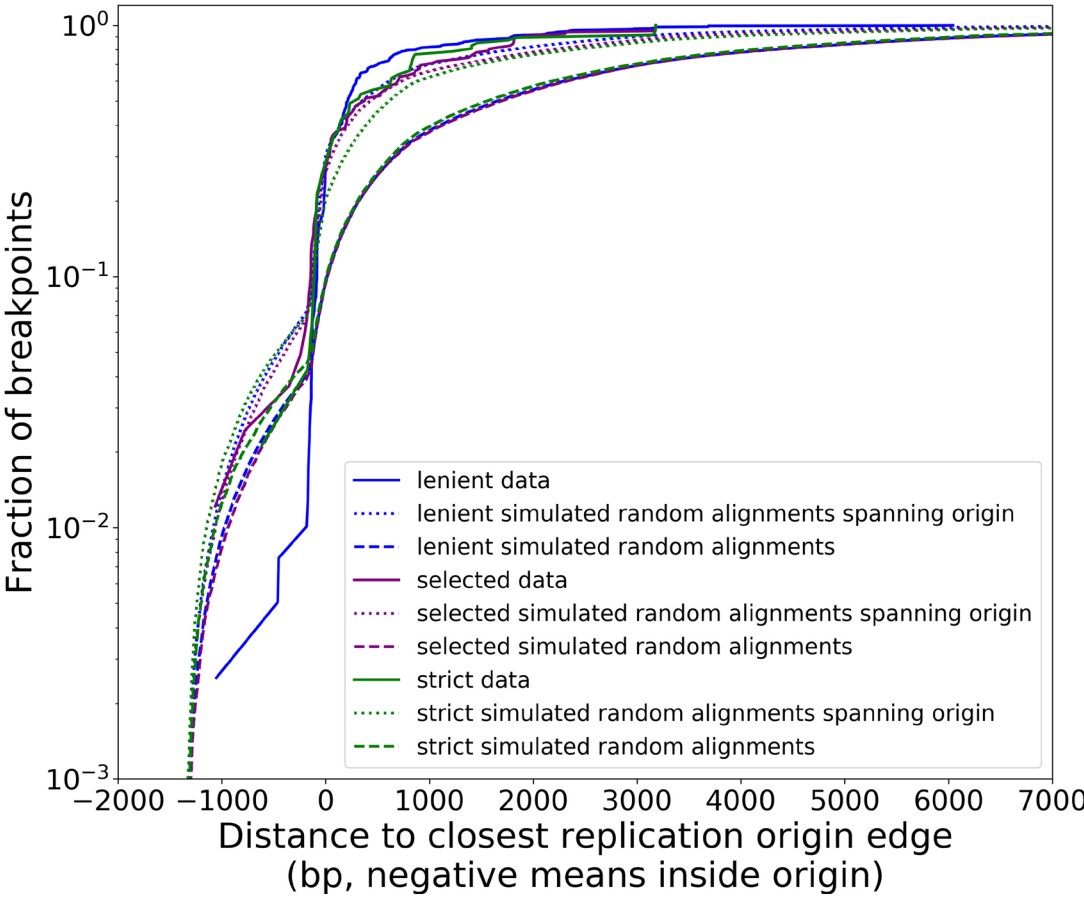

**Appendix 1—figure 3.** Distribution of minimum distances between alignment edges and origin edges is robust to different structural detection pipeline parameter regimes. Real data (solid lines in this plot) follow all filtering steps in our structural pipeline, including inverted duplication artifact filtering, and majority voting (except 'lenient' parameter set). The different data curves are a result of the aforementioned filtering steps in the pipeline, but with different parameters (*Appendix 2—table 2*). Small, dotted lines represent simulations of uniform random alignments spanning origins, with length distributions of alignments from each set of data curves. Long dashed lines represent the same type of simulation with no requirement for alignments to span origins. Both the green and purple data curves reside close to the 'strong origin selection' models or the small, dotted lines. The blue parameter regime, which we would expect to cluster more noise because we are being more lenient with filtering thresholds, differs at least to a larger degree than green/purple. Overall, however, all three parameter regimes perform similarly, suggesting that the shape of these distributions and the claims we are making here are robust at least to changes in parameter values in our structure detection pipeline. In another sense, this suggests that minimap2 is already neglecting most base-level changes very well and only considering severe deviations in expected collinearity to be the end of alignments that form breakpoints.

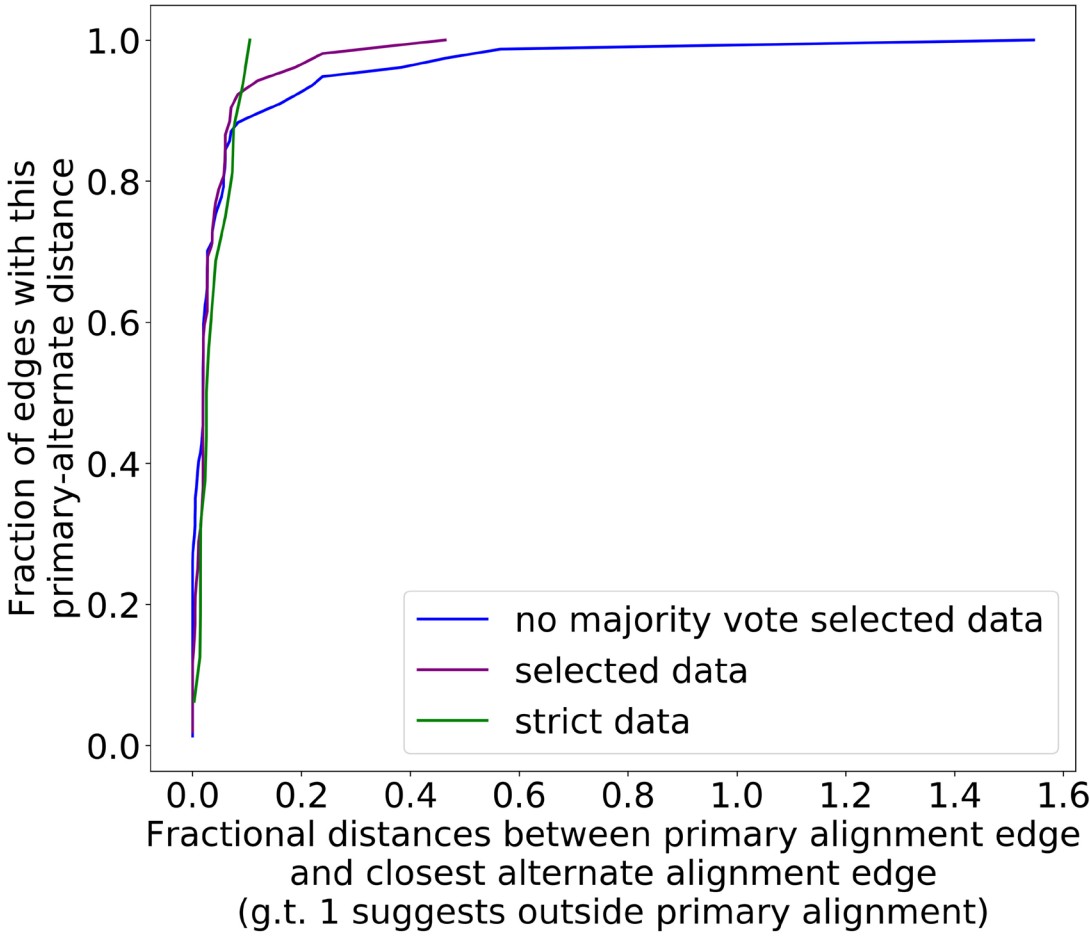

**Appendix 1—figure 4.** The preference for primary structure excision sites is robust to different filtering parameter regimes. Here, we are plotting the cumulative distribution of fractional distances between the primary alignment edge and closest alternate alignment edge for three different parameter regimes (*Appendix 2—table 2*). The blue curve, which represents our 'lenient' regime in this case, is now simply the purple curve (parameters in *Appendix 2—table 1*, regimes in *Appendix 2—table 2*) without majority voting. This change is necessary compared to *Appendix 1—figure 3* because we now must infer structure through repeat detection, which requires more strict parameters to begin with. Without the majority voting it is clear that alternate alignments begin to creep outside primary alignments due to noise, but as a whole, all three parameter regimes display a high density at small fractional distances, suggesting a preference for excision sites across or near the primary structure excision site as described in the main text.

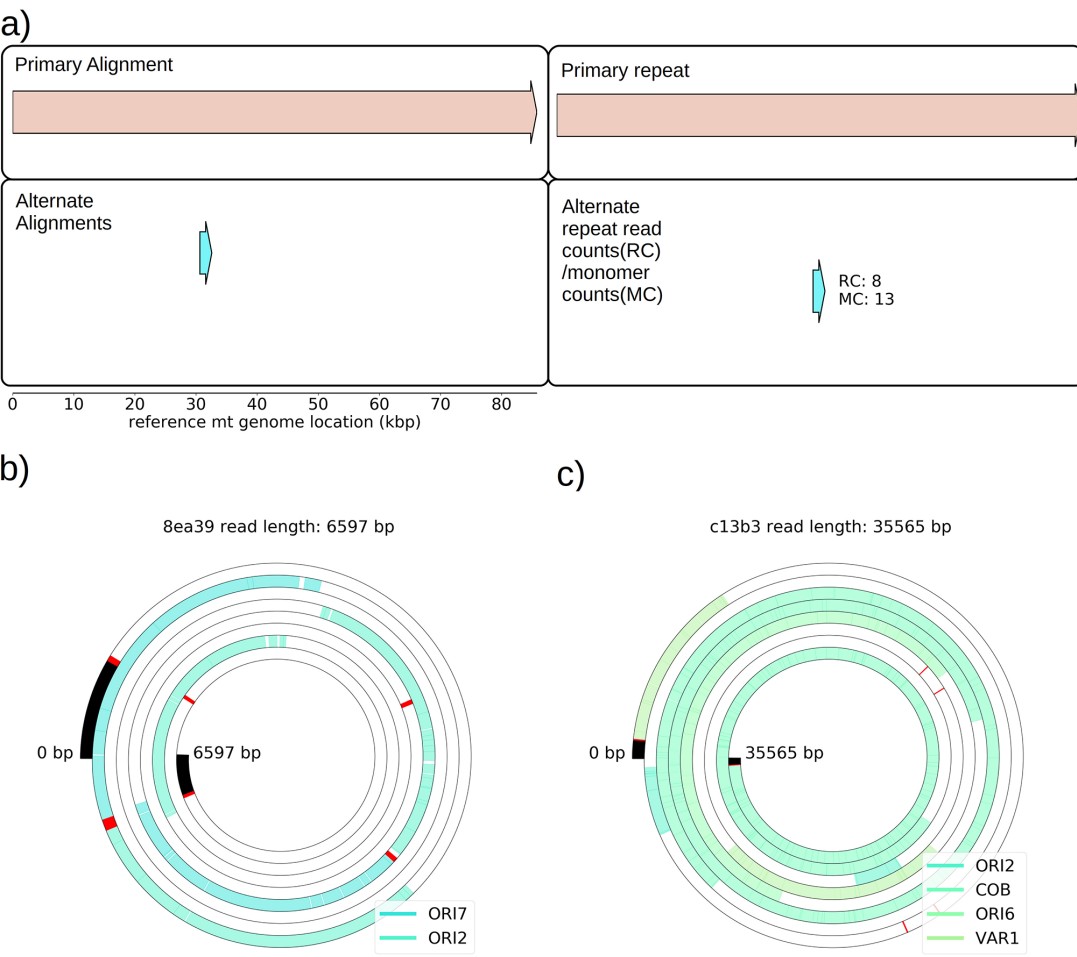

**Appendix 1—figure 5.** Structures observed in Grande cells grown in YPG—evidence of heteroplasmy. (**a**) The first panel on the left shows the only high confidence primary and alternate structure detected in one of the four YPG Grande samples sequenced. Primary structure is an intact genome, while the alternate structure is a repeat that spans two origins of replication near 30 kbp. The panel on the right shows the numbers of reads that support this alternate structure (RC), and the estimated number of monomer (repeat unit) counts observed across reads (MC). Given that true Petite lineages cannot exist in YPG media, this signal is due to at least transient heteroplasmy within cells. While this is the only structure detected across four samples, there are other breakpoints present at low frequencies in other YPG Grande samples, but they are not prevalent enough to infer high confidence structures from. Structure detection from Grandes is difficult because these structures are diverse (afforded by the WT genome being the reference point) and exist within cells and not lineages (therefore not enriched by chance through early bottlenecks). Given the results from *Marotta et al., 1982* that show this ori2–ori7 breakpoint is most prevalent in Petites, it is not surprising that it was only this breakpoint that was detected in YPG Grandes. (**b**) An example read contributing to this breakpoint signal. Annotated features are colored and labeled blocks, red lines are breakpoint locations determined from mapping. (**c**) An example of another structure in this same YPG sample that has been detected but is not prevalent enough for the algorithm to infer its structure (require >3 read support, and more than two periods of a repeat). It is a direct repeat of a region spanning from ori2 to the var1 gene.

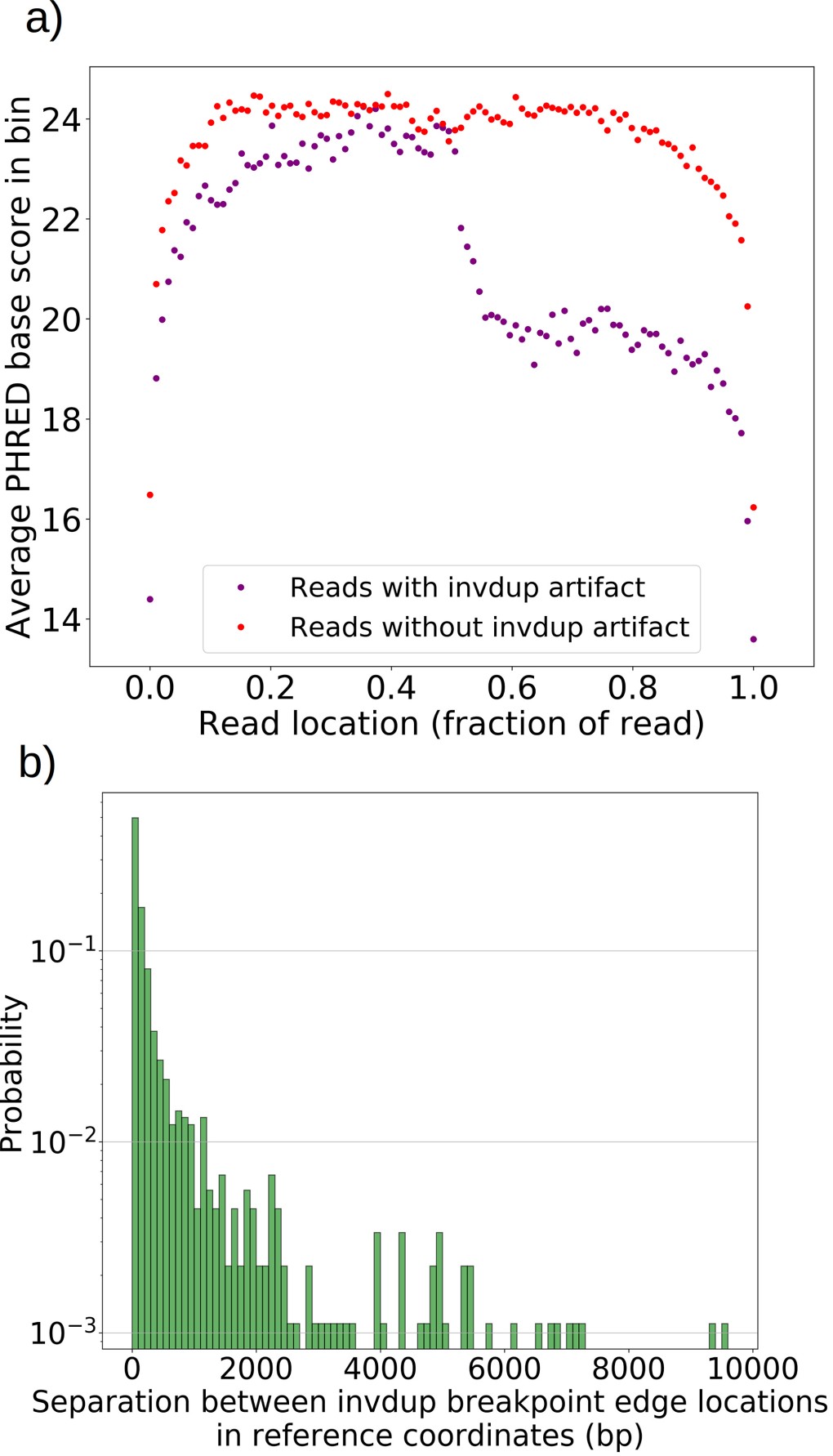

**Appendix 1—figure 6.** Inverted duplication artifact breakpoint edges are further separated than expected due to reduced base calling accuracy. (**a**) For all reads with and without invdup artifacts in chrM in one representative Grande sample, a sliding window of 100 bp was moved across the reads and the average PHRED scaled base quality score was computed in this window. Reads with invdup artifacts have a clear decrease in PHRED quality score in the latter half of the read due to complementary strand interaction and ratcheting as described in *Spealman et al., 2020*. This decrease in base call quality means that mapping algorithms have to contend with more noise, resulting in a larger separation distance between invdup breakpoint edges which in an ideal case would perfectly coincide in reference coordinates. (**b**) The distribution of separation distances between invdup breakpoint edges in reference coordinates. Breakpoint edges largely (90%) reside within 1 kbp of each other in reference coordinates. Inverted duplications residing outside of the expected artifact read location distribution and beyond 1 kbp separation are unlikely to be this particular sequencing artifact. We rely on these ideas in the filtering described in Materials and methods.

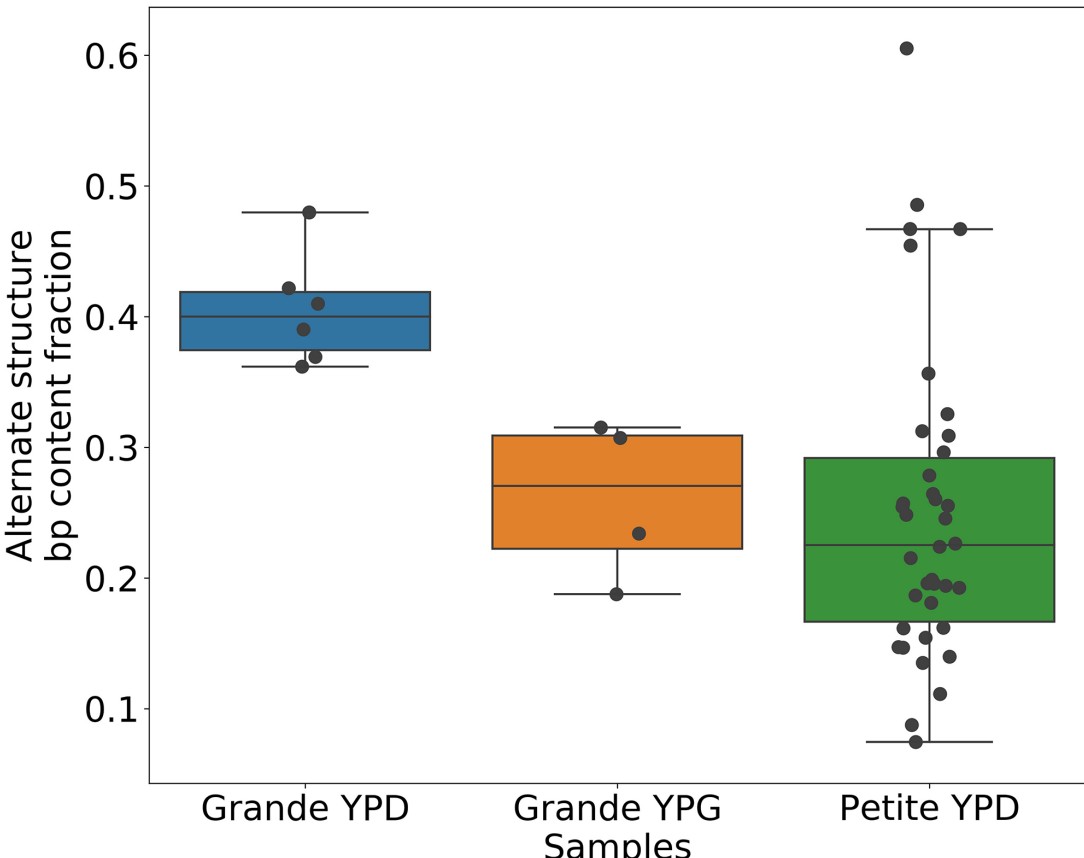

**Appendix 1—figure 7.** Alternate structure frequencies when not neglecting reads containing inverted duplication artifacts. In the above plot, we are computing the base-pair content fraction of any reads that contain a breakpoint we have deemed to not be an inverted-duplication artifact, regardless of whether or not an artifact is present in the read. Each dot represents this alternate structure fraction across all reads in a single strain sequenced. The box plot displays the minimum, maximum, first quartile, and third quartile. We have done this same calculation for Grande samples in YPD/YPG media, and in Petites. Given that YPG samples have 2× the number of inverted duplication artifacts, which are often accompanied by spurious breakpoints, the discrepancy here between YPD and YPG samples is strongly suggestive of clonal divergence playing a role.

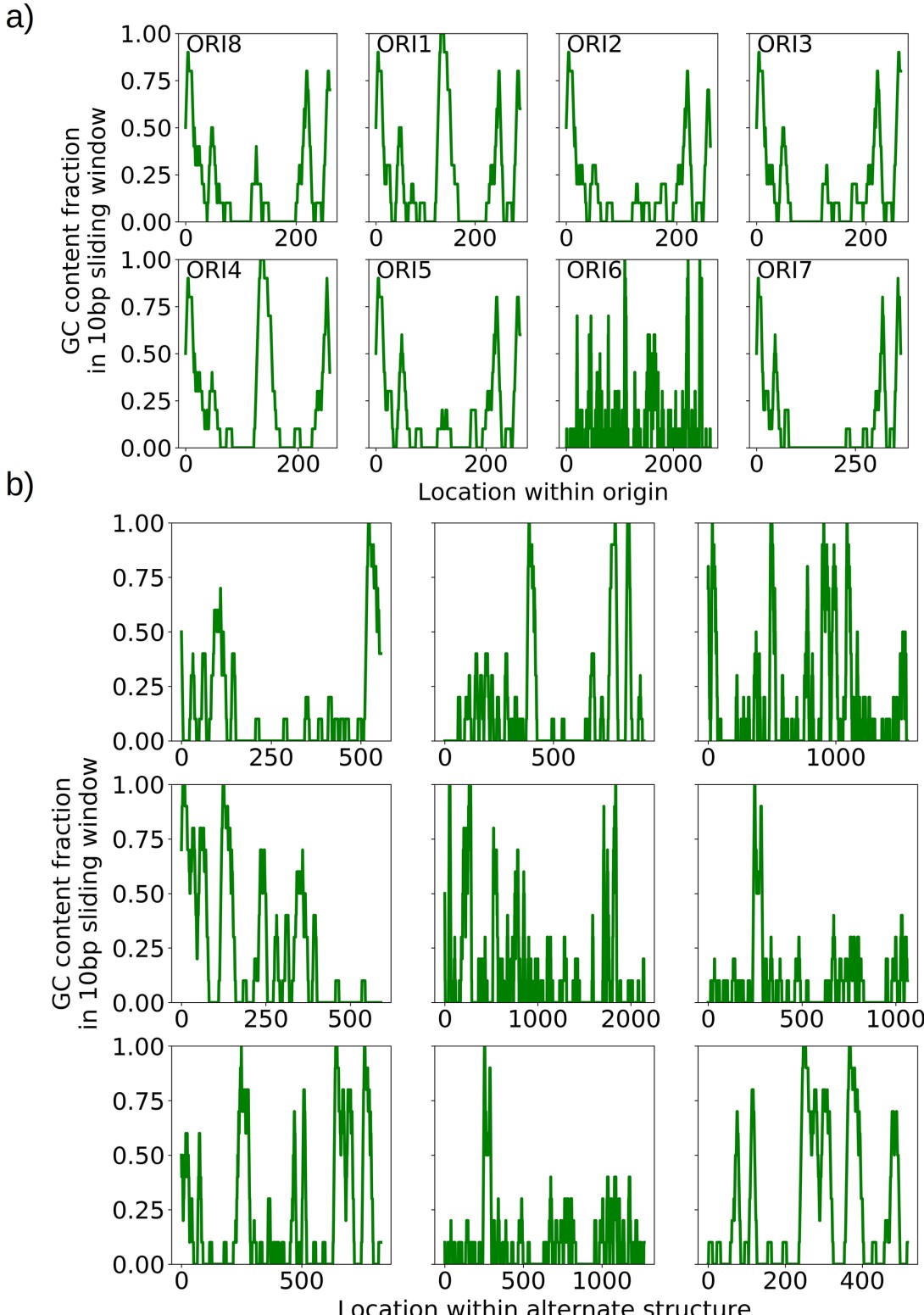

**Appendix 1—figure 8.** GC clusters in origins and alternate structures without origins. (**a**) A visualization of the distinct GC clusters within all eight origins of replication. A 10 bp sliding window is moved along the origin sequences. GC clusters are the 3–4 distinct peaks in each of the origin sequences, all above 0.6 GC content in the sliding window which are consistent with the locations described in *de Zamaroczy and Bernardi, 1986*. (**b**) A visualization of the GC clusters within observed alternate structures without canonical origins of replication. These clusters at similar GC content (0.6 and above) may act as surrogate replication origin sites.

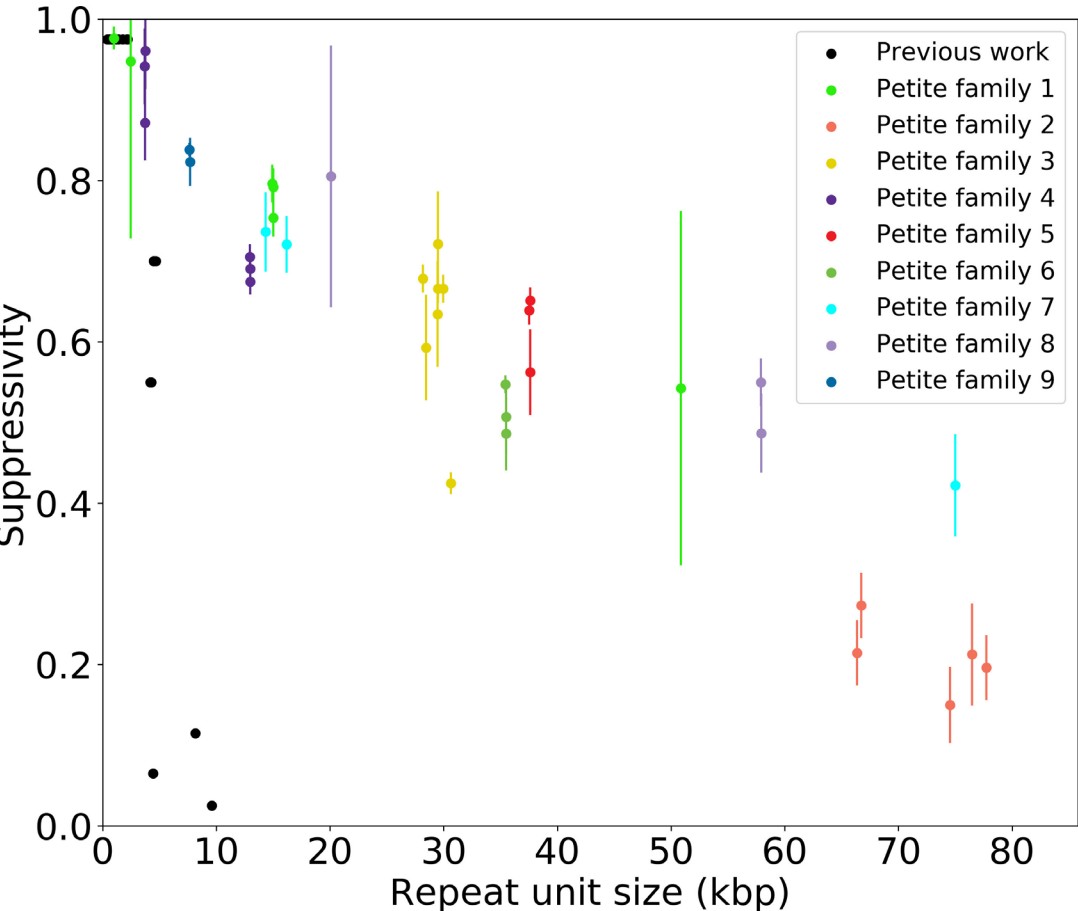

**Appendix 1—figure 9.** Supressivity and its correlation with repeat unit size. Black dots represent the average suppressivity and repeat unit size from the range of suppressivities and repeat unit sizes published in *de Zamaroczy et al., 1981*; *Mangin et al., 1983*. Each colored dot indicates the average suppressivity across three second passage Petites derived from the same first passage progenitor. Y-axis error bars are ± the standard deviation in suppresivities across these three second passage Petite colonies. Shared colors indicate that these strains were derived from the same spontaneous Petite colony and constitute a Petite family. As in the main text, repeat unit size is taken to be the sum of the unique alignment lengths in the primary alignment of each sample. Samples containing inverted breakpoints in their primary alignments are those in families 2 and 3, the orange and yellow dots, respectively. Family 3 was confirmed to be a mixed sample.

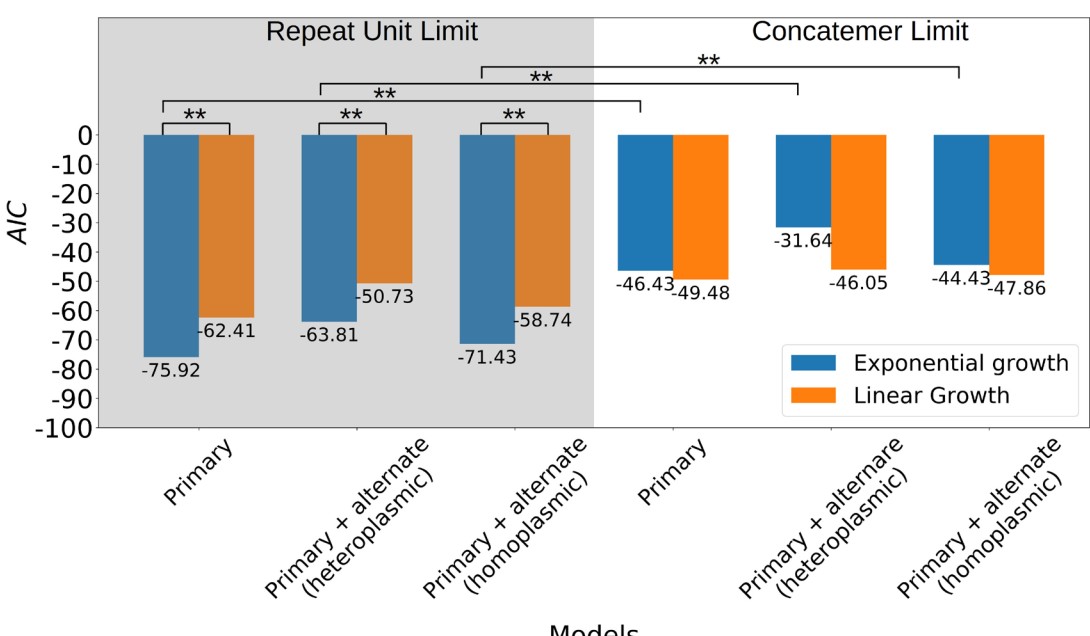

**Appendix 1—figure 10.** Comparison of suppressivity models. The y-axis is the Akaike Information Criterion, computed from a least-squares fit for each model by transforming the least-squares statistic into a Normal negative log-likelihood statistic. The left pane includes models that exist in the repeat unit limit, meaning growth rate terms ($\rho_i \nu_i$) either in the exponent (exponential), or as is (linear) are multiplied by the inverse repeat unit length ($1/L_i$), where is Petite or Grande. These products represent the time evolution of the abundance of each structure ($N_i$ as shown in the axis of *Figure 8* in the main text). The right pane is the concatemer limit, which takes the same form of the models in the left except with the exclusion of the inverse repeat unit length prefactors. The notation 'Primary' indicates that only primary structures are considered in the theoretical suppressivity calculation, while 'Primary+alternate' indicates that both alternate structures and primary structures observed in strains contribute to the theoretical suppressivity. There are two ways alternate structures are included in the models here: (I) The heteroplasmic limit, where it is assumed that these structures coexist. In this case, the relative contributions of alternate/primary structures are included by computing $N_P$ as an average over all structures weighted by their mitochondrial contributions. (II) The homoplasmic limit, where it is assumed that all structures are segregated into their own homoplasmic lineages. In this case, the theoretical suppressivities are an average weighted by mitochondrial contributions of each structure. By comparing the relative likelihood of each model, in the repeat unit limit the exponential model is significantly favored over the linear model (**, $2\sigma$). All pairwise comparisons between the same models in the repeat unit limit are significantly favored over the concatemer models with the exception of the heteroplasmic linear models. Homoplasmic/heteroplasmic limits with the inclusion of alternate structures have little effect.

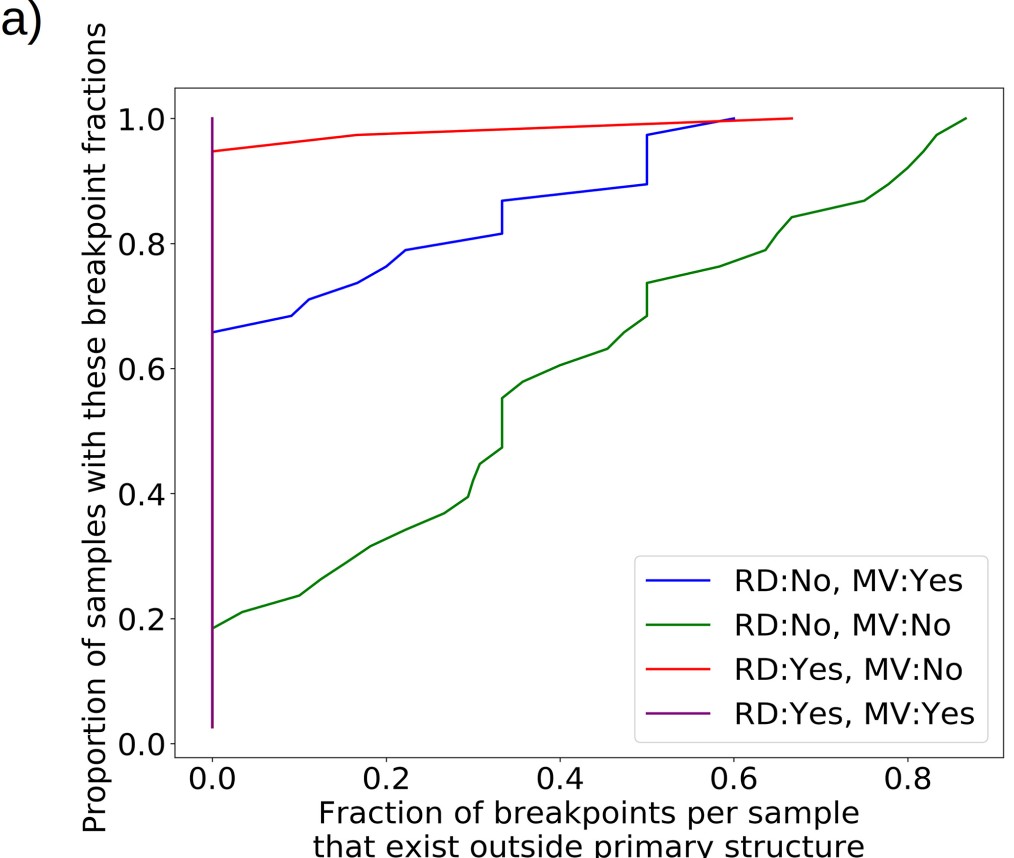

a)

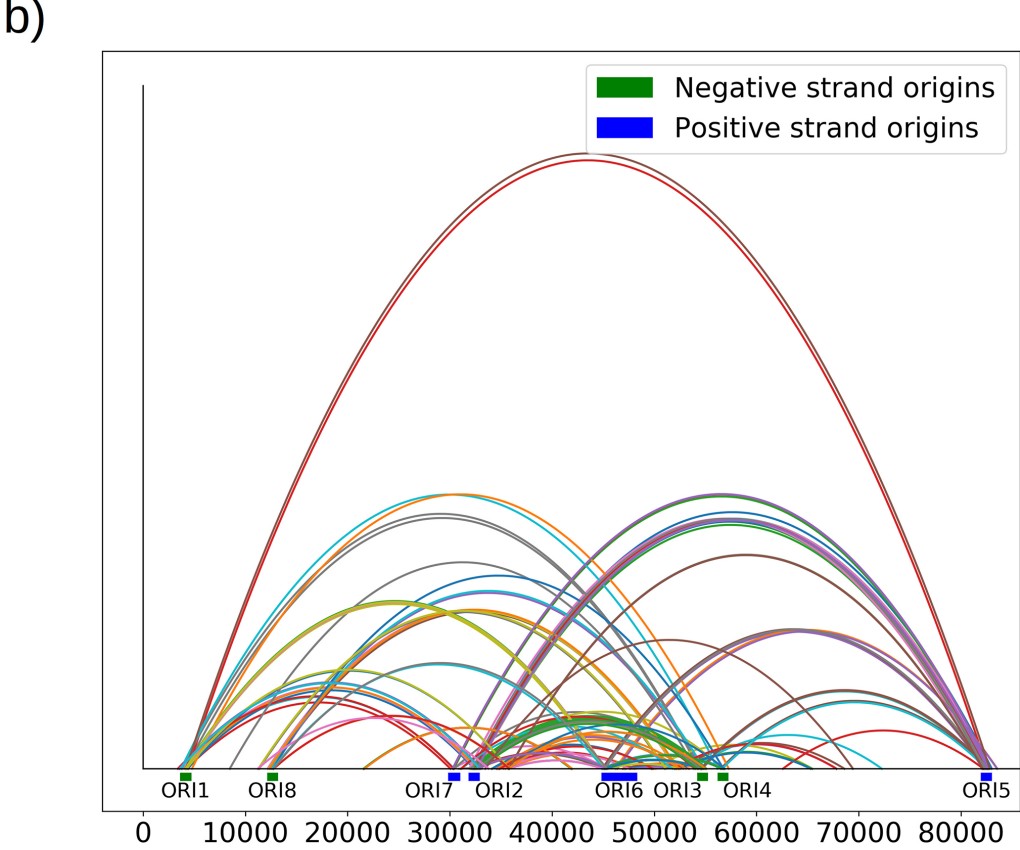

b)

**Appendix 1—figure 11.** Effect of repeat detection (RD) and majority voting scheme (MV) on breakpoint filtering. (**a**) Between the green and blue curves, we see the effect that majority voting (MV in legend) has on the cumulative fraction of breakpoints that exist outside the primary alignment (and are therefore unlikely to be real given the notion of an excision cascade). Between the green and red curves, we see the effect of tandem repeat detection (RD in legend) of breakpoint labels in reads, which largely eliminates breakpoints that exist outside of the primary alignment. The purple curve indicates the effect of both repeat detection and majority voting, indicating that across all samples breakpoints are contained within the primary alignment which makes them believable. (**b**) Here, we are plotting the locations of breakpoints that are removed through repeat detection and the majority voting process. Breakpoints that are removed this way largely bridge between origins of replication due to significant homology. Sequencing errors can perturb one origin into another, resulting in these spurious breakpoints that are removed by these two schemes.

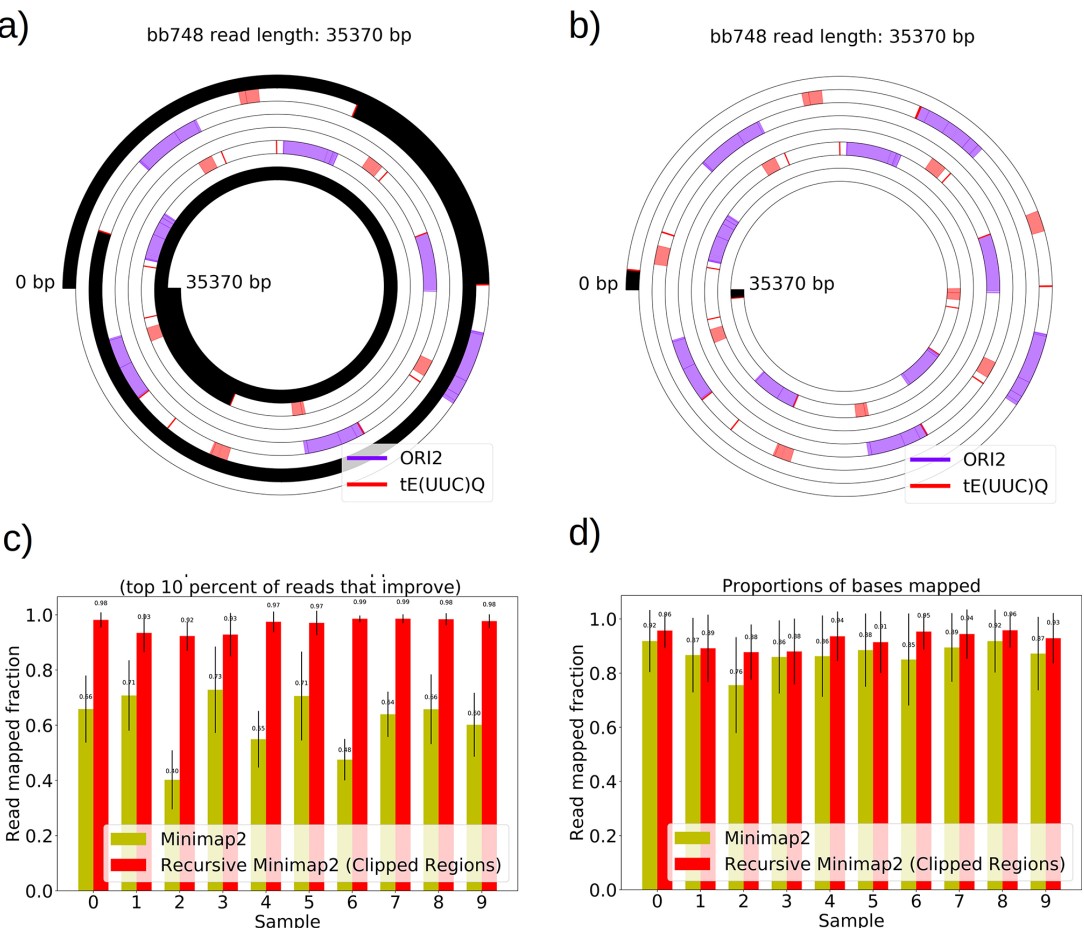

**Appendix 1—figure 12.** The necessity and effect of recursive Minimap2 alignments in resolving Petite structure. (**a**) A spiral plot showing a raw Minimap2 alignment of a Nanopore read with default parameters. The legend indicates annotated features that are present, black regions represent clipped (unmapped) regions, red bars indicate the start or ends of adjacent alignments. Large portions of this read are unmapped due to the default Minimap2 z-offset parameter, which truncates repeated alignments due to the expectation of colinearity with the reference sequence. Instead of varying this parameter, which requires balancing early truncation and enforcing colinearity with the reference, we opted to recursively apply Minimap2 in unmapped portions after the first run. (**b**) The effect of recursively mapping unmapped portions in the same read which almost entirely eliminates the unmapped regions except at the ends of reads where adapters still reside, and sequencing error is generally higher. While this produces a pseudo-global alignment, only alignments with MAPQ>20 are retained in subsequent analysis so our requirements of alignment specificity are maintained. (**c**) The most impactful change that recursive mapping has in improving mapping fraction in 10 samples. (**d**) The global effect, which is minimal, but does improve statistics in repeat detection and structure construction especially in low frequency structures where every read counts.

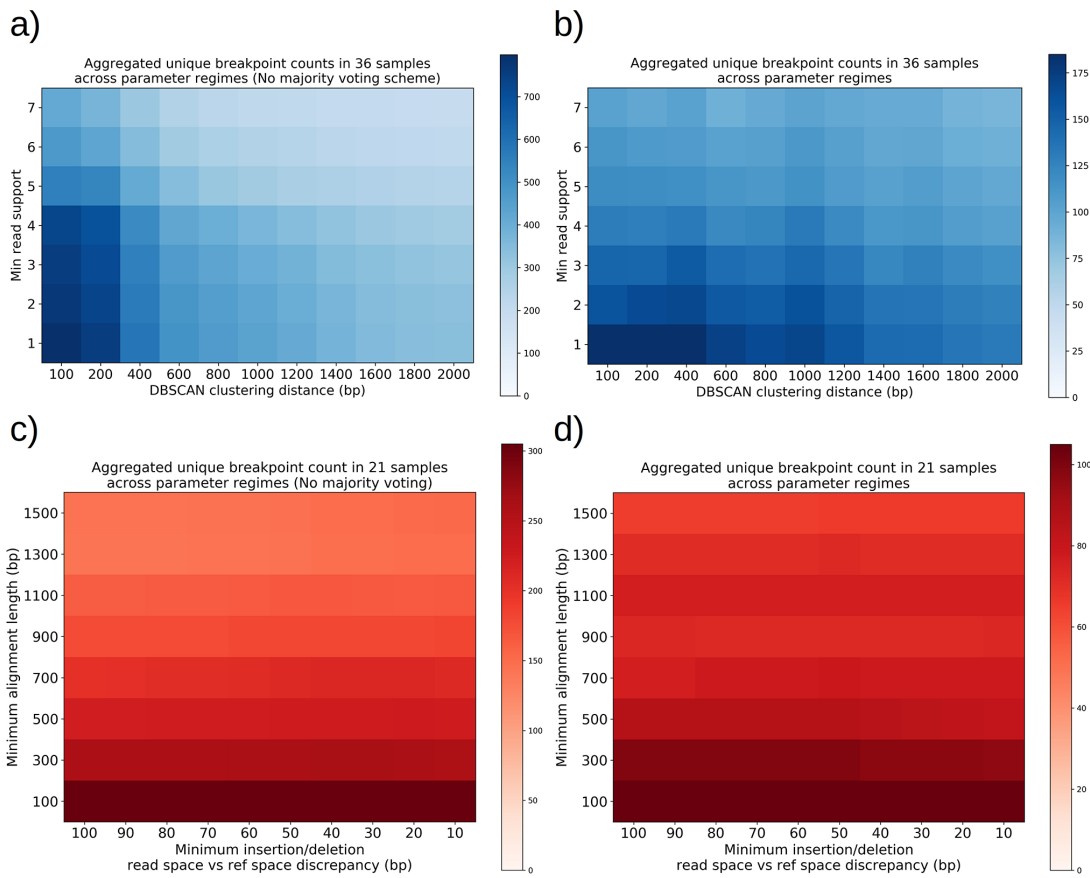

**Appendix 1—figure 13.** Structural detection pipeline parameter sweeps—The effect of majority voting, minimum read support, and DBSCAN clustering radius, insertion/deletion size threshold, and minimum alignment length. (**a**) A plot of total unique breakpoint counts (identified through DBSCAN clustering) without applying the majority voting scheme described inMaterials and methods across a parameter sweep of DBSCAN clustering radius and minimum read support. Minimum alignment length and minimum insertion/deletion size were fixed at 300 and 30 bp, respectively. As expected, we see a monotonic decrease in counts as clustering radius is expanded, and the same type of decrease for increasing read support. The high-density cluster to the lower left is an indication that in this regime, we are largely clustering noise. (**b**) The same plot in (**a**) but with majority voting for breakpoints, which results in a fourfold decrease in unique breakpoint counts at the extremum. The flatness across these parameter ranges is a sign that the breakpoint counts represent real structures that are largely insensitive to parameter selection with the exception of a small clustering radius which will always produce more clusters in the lower left. (**c**) A plot of total unique breakpoint counts (identified through DBSCAN clustering) without applying the majority voting scheme described in Materials and methods across a parameter sweep of minimum insertion/ deletion length and minimum alignment length. The DBSCAN clustering radius and minimum read support were fixed at 1 kbp and 3, respectively. We see low variance in counts along the insertion/deletion length threshold axis, suggesting that most breakpoint calls from minimap2 on its own are capturing real breakpoints and that this parameter has little effect. Increasing minimum alignment length as expected results in a monotonic decrease in breakpoint counts because less reads exist in the tail of the read-length distribution, and because small structures are thrown away. (**d**) The same plot in (**c**) but with majority voting for breakpoints, which results in a threefold decrease in unique breakpoint counts at the extremum. While we see less steep descent along the minimum alignment length axis, it is clear that below 300 bp in alignment length there appears to be a transition to clustering of noise, which is due to spurious replication origin to replication origin transitions (*Appendix 1—figure 11*).

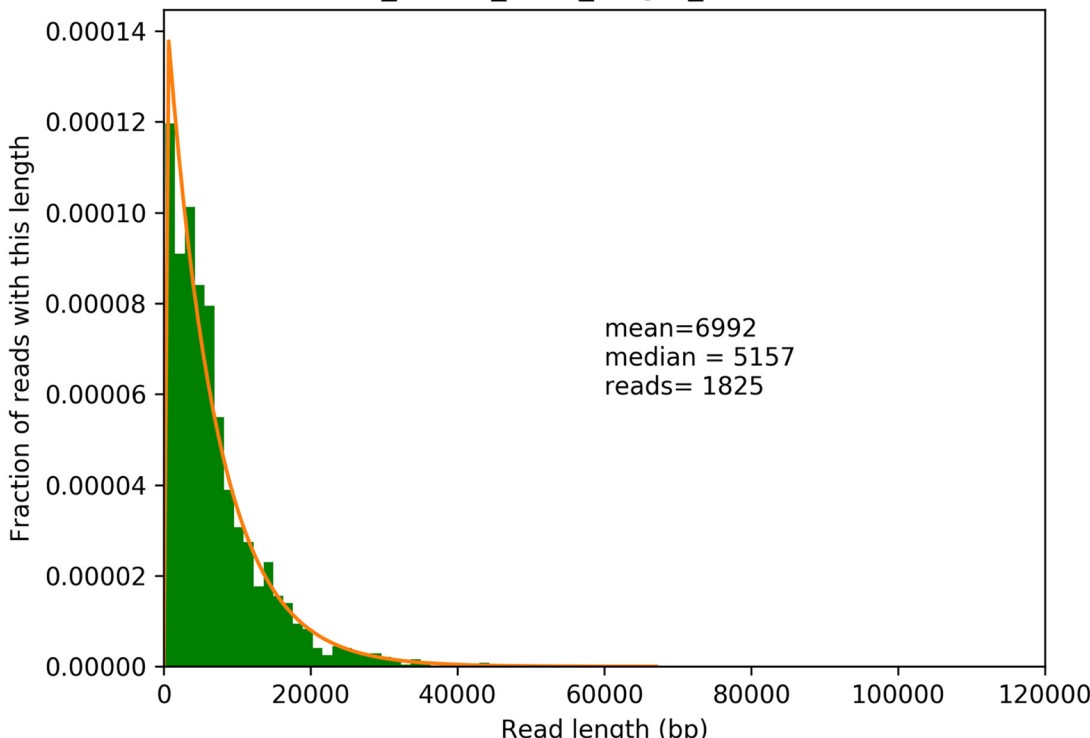

**Appendix 1—figure 14.** Example read length distribution from sequencing of a single Petite strain and its exponential fit. A plot of an empirical read length distribution from one sequence Petite strain. Mean, median, and the number of reads are denoted in addition to a fit to an exponential probability distribution in orange. For this particular fit, the location and scale parameters were 213 and 6779, respectively.

# Appendix 2

## Additional Tables

### Summary of algorithm

1. Data set across all samples is restricted to reads that pass default Guppy QC, alignments with PHRED alignment score >20 and alignment length >300 bp.
2. Breakpoints in alignments where the read-space and reference-space differ by >30 bp are recorded as 'breakpoints.'
3. Inverted breakpoints, where the strand changes mid read, are filtered from hairpin artifacts based on the positions within reads, allowing for 1% false positive error at this stage (see *Appendix 1—figure 1*). Hairpin artifacts are marked for use later in a majority voting scheme.
4. Non-inverted breakpoints, and inverted breakpoints are separately clustered using DBSCAN, with a clustering distance of (eps) of 1 kbp, and a minimum cluster occupancy of 3. K-means clustering, with $k$=2, is also performed within clusters in case breakpoint transitions occur within a cluster, indicating too coarse DBSCAN clustering. This happens for example in a 1LH to 1HL encoded breakpoint transition, although in practice it was rare in the data (3/35 alternate structures, and no primary structures required subsequent clustering).
5. All breakpoints are further filtered with a majority voting scheme which eliminates clusters that exist due to sequencing error and are not periodic (assuming concatemer structures), or are not recapitulated on either strand of read when both strands are present. The latter is reminiscent of Oxford Nanopore 2D basecalling, but is performed post alignment.
6. Breakpoints are further filtered so that only those represented across >3 separate reads remain.

**Appendix 2—table 1.** Structural detection pipeline overview and summary of parameters. Detailed summary of parameters.

| Parameter | Value | Description/justification of choice of value |
|---|---|---|
| PHRED alignment score | 20 | 1/100 probability of a false positive alignment by chance given a particular size/complexity of reference |
| Minimum alignment length | 300 bp | Mt replication origins are ~300 bp on average and exhibit significant homology. Below 300 bp, alignments containing origin fragments are often indistinguishable in Nanopore sequencing error background, resulting in erroneous alignments that break expected collinearity. This would also be the expected lower limit for detectable repeat units in Petites. See *Appendix 1—figure 13* for effects of this parameter. |
| Insertion/deletion threshold | 30 bp | The minimum discrepancy in reference ersus read space to call a breakpoint. This parameter appears to have little effect at low values, suggesting breakpoints we see break collinearity by much larger distances, and are therefore more believable over Nanopore sequencing error background which often introduces small (~10 bp) insertions and deletions. See *Appendix 1—figure 13* for effects of this parameter. |
| DBSCAN epsilon (minimum clustering distance) | 1 kbp | This is the upper threshold for Sniffles (*Sedlazeck et al., 2018*) and NanoSV (*Cretu Stancu et al., 2017*) clustering, and will cluster within smaller distances if the SV's themselves are smaller. Varying this value has little effect even in a range of a few hundred bp due to the majority voting scheme. See *Appendix 1—figure 13* for the effect of this parameter. |
| DBSCAN minimum cluster occupancy | 3 | Fixed at 3 in this pipeline to allow for small clusters, and generally affected by coverage. More stringent filtering along similar lines is available with read support filtering. Also important to note that we do not see high confidence structures with breakpoint counts <10 once repeats have been detected. |
| Minimum read support | 3 | Reasonable values for this parameter are largely dependent on sequencing coverage. See *Appendix 1—figure 13*. |

**Appendix 2—table 2.** Definitions of parameter regimes referenced in *Appendix 1—figure 3* and *Appendix 1—figure 4*.

| Parameter | 'Lenient' parameter set value | 'Selected' parameter set value | 'Strict' parameter set value |
|---|---|---|---|
| Minimum alignment length | 100 bp | 300 bp | 1 kbp |
| Insertion/deletion threshold | 30 bp | 30 bp | 30 bp |
| DBSCAN epsilon (minimum clustering distance) | 100 bp | 1 kbp | 1.4 kbp |
| Minimum read support | 1 | 3 | 5 |
| Majority voting? | No | Yes | Yes |

