## [Editor Report]

This work provides further insight into long-time outstanding questions in the field of mitochondrial genetics using long-read sequence analysis and biophysical modeling. Specifically, the authors show how replication origins in mitochondrial DNA are recombination hotspots that can result in excision cascades, that lead to a variety of different mitochondrial mutants and in some cases even heteroplasmic cells. Finally, crossing wild-type cells with various mitochondrial mutants allowed the development of a model for the suppressivity (fitness) of different mitochondrial variants that suggests that the density of replication origins in different repeated units is a major determinant of mtDNA suppressivity.

---

## [Decision Letter]

**Decision letter after peer review:**

Thank you for submitting your article "Contingency and selection in mitochondrial genome dynamics" for consideration by *eLife*. Your article has been reviewed by 3 peer reviewers, including Kevin J Verstrepen as Reviewer #1 and Reviewing Editor, and the evaluation has been overseen by Naama Barkai as the Senior Editor. The following individual involved in review of your submission has agreed to reveal their identity: Konstantin Khrapko (Reviewer #3).

All three reviewers essentially reach the same conclusion, namely that your study's use of today's high-throughput long- read length technology yielded a valuable comprehensive view of mt DNA dynamics in Saccharomyces. However, we all feel that the work falls short when it comes to clear framing of the results in a broader context. Hence, we feel that a thorough revision of the introduction and Discussion section is needed to better summarize the existing literature in a concise and accessible way.

Essential revisions:

1. Please clearly listing the existing knowledge and open questions and which of these are confirmed, revised or answered with this new study.

2. In the Discussion section, please frame the results more broadly, and discuss the parallels and differences to other organisms, including humans and human mitochondrial diseases.

3. Please consider the more detailed comments in the each of the reviewer's comments below and adapt the text accordingly, or provide an argument why changes are not needed.

*Reviewer #1 (Recommendations for the authors):*

Specific questions and concerns

Note that the text does not contain line numbers, making it difficult to pinpoint specific sections…

Overall, the text is long and wordy, and not always clearly written, especially for non-experts. It would be advisable to rework the introduction section, and better summarize and explain existing literature, as well and clearly define principles like "suppression". It seems that the most relevant existing body of literature can be summarized in a much more compact text, that stresses the existing hypothesis that the formation of petite mutants is driven by the presence in the mitochondrial DNA of short repeats that favor recombination events, followed by the selection of (nonfunctional) fragments that contain a large density of replication origins, causing these fragments to be replicated more efficiently than the larger, complete mt genome. The selection for a higher density of origins also favors the formation of complex concatemers of mtDNA fragments that contain multiple origins. When this is summarized in the first paragraph of the introduction, it becomes much easier for the reader to understand the more specific questions about the formation and selection of mt fragments and the questions that remain unanswered.

Page 7: "Whether this theory of origin-origin recombination, or preferred excisions near replication origins holds up, we were curious to understand the role replication origins might play in the excision process." ◊ revise sentence

Figure 3, legend – please provide a title for the figure, and make sure that the first line for each panel describes what is in the panel. (this is OK for panel b, but not for a and c).

Figure 3, panel B: why does the green curve have such a strong inflection point near zero? Why is it not symmetrical around zero?

Paragraph on heterplasmy – I am not sure that I understand the arguments for the presence of heterplasmy. This seems very indirect. Please explain in a more clear way. Also, why not simply do single-cell sequencing?

Discussion section regarding the outliers in repeat size – is it also possible that some outliers might simply reflect that selection is still in progress? Not all structures with sub-optimal fitness disappear instantly – selection often takes some time (ie replication cycles).

Discussion, last paragraph: I do not agree with the conclusion that these findings suggest that feral yeasts have maintained their mtDNA as a result of continuous selection for respiratory metabolism. It would be useful to consider and discuss the fact that feral yeasts generally have much lower petite formation; and consider if and how the use of a laboratory strain like W303 influences the results.

Note that laboratory strains and some beer yeasts likely lost the mechanisms that reduce mt DNA instability because they have been relieved from selection for respiratory metabolism for many years of repeated growth under fermentative conditions in laboratory medium (high glucose, inducing Crabtree repression); but feral yeasts do not need continuous selection for respiration to maintain their mtDNA.

*Reviewer #2 (Recommendations for the authors):*

1. In Figure 2b and 2c, having the entire schematic of the yeast mitochondrial genomes rather than just sub-sections would make your point about the repeat concatemer structure found Petite genomes while also making the work more accessible by those who don't use yeast as their model organism.

2. Additionally, please move the legend in figure 2c off the figure. Also, increasing the visibility of the red breakpoint locations (especially in the Grande genome illustration) will allow readers to better compare the structural differences in the genomes.

3. The use of the x,y plot for figure 2d is a bit confusing. Mapping all the breakpoint locations to schematic of the yeast mtDNA genome might more clearly illustrate the greater quantity and distribution of breakpoints found in the Petite genomes as compared to the Grande genomes.

4. In the Discussion section, citing other systems that this sequencing and structural inference system could be applied to would help to increase the relevance of this system outside of your specific research question.

*Reviewer #3 (Recommendations for the authors):*

It is imperative that all data is made available to the reader in the raw and processed format, with annotation sufficient to repeat analyses and to run novel analyses. At this point I do not see clear data availability section.

---

## [Author Response]

Essential revisions:1. Please clearly listing the existing knowledge and open questions and which of these are confirmed, revised or answered with this new study.

We have addressed this point directly in the rewritten introduction and Discussion section. The introduction section now concisely covers the existing knowledge and specifically lists the questions we are addressing (Line # 29-121). The same structure is also mirrored in the discussion, where we summarize the answers to these questions and existing knowledge that applied to them (Line # 674-857).

2. In the Discussion section, please frame the results more broadly, and discuss the parallels and differences to other organisms, including humans and human mitochondrial diseases.

The two final paragraphs of the Discussion section now conclude with parallels between Petite mtDNA dynamics and mtDNA dynamics in humans (Line # 839-857). The abstract has also been updated to reflect this change in framing (Line # 9-22).

Reviewer #1 (Recommendations for the authors):Specific questions and concernsNote that the text does not contain line numbers, making it difficult to pinpoint specific sections…

The article file now contains line numbers which are referred to in this response.

Overall, the text is long and wordy, and not always clearly written, especially for non-experts. It would be advisable to rework the introduction section, and better summarize and explain existing literature, as well and clearly define principles like "suppression". It seems that the most relevant existing body of literature can be summarized in a much more compact text, that stresses the existing hypothesis that the formation of petite mutants is driven by the presence in the mitochondrial DNA of short repeats that favor recombination events, followed by the selection of (nonfunctional) fragments that contain a large density of replication origins, causing these fragments to be replicated more efficiently than the larger, complete mt genome. The selection for a higher density of origins also favors the formation of complex concatemers of mtDNA fragments that contain multiple origins. When this is summarized in the first paragraph of the introduction, it becomes much easier for the reader to understand the more specific questions about the formation and selection of mt fragments and the questions that remain unanswered.

We thank the reviewer for this suggestion. We have rewritten the introduction section so that it starts with a concise summary of existing work, followed by the questions we are exploring, and what existing knowledge pertains to these questions (Line # 29-121).

Page 7: "Whether this theory of origin-origin recombination, or preferred excisions near replication origins holds up, we were curious to understand the role replication origins might play in the excision process." revise sentence

A more clear and concise description for the motivation for investigating origin-excision proximity is now present in the manuscript in the corresponding Results section (Line # 233-235). We thank the reviewer for this suggestion.

Figure 3, legend – please provide a title for the figure, and make sure that the first line for each panel describes what is in the panel. (this is OK for panel b, but not for a and c).

Figure titles have been added to all figures where they were missing, and panel descriptions are now complete in Figure 3. We thank the reviewer for this comment.

Figure 3, panel B: why does the green curve have such a strong inflection point near zero? Why is it not symmetrical around zero?

The inflection point in the green curve just below zero is due to the size of the replication origins in the mtDNA reference. This dependence on replication origin size is also why random alignment models in the black and orange curve follow this trend.

Seven of eight replication origins are ~300bp in size, and one is ~2700bp. For breakpoints that exist inside the origins, meaning below zero in this plot, the majority exist in the displacement range from -150bp to 0bp. This range corresponds to the minimum displacement between a breakpoint inside a 300bp origin and the origin’s edge. Because most origins are 300bp, and breakpoint displacements are restricted to -150bp to 0bp inside them, this is the reason for the inflection point below zero. Regarding the asymmetry around zero, once we move beyond the confines of breakpoints in origins (to +ve values), breakpoints can be as far away as origins are separated in the mtDNA reference. This is why we see multiple kbp of displacements for +ve values on the x-axis. We thank the reviewer for this detailed observation.

Paragraph on heterplasmy – I am not sure that I understand the arguments for the presence of heterplasmy. This seems very indirect. Please explain in a more clear way. Also, why not simply do single-cell sequencing?

We thank the reviewer for this comment which we agree warrants both discussion here and changes to increase clarity in the corresponding Results section, which we have implemented. The changes are present in the results subsection How is the observed structural heterogeneity of mtDNA in yeast colonies partitioned among individual cells? (Line # 498-550). We take this comment to be referring to the homoplasmy/heteroplasmy paragraphs on Petite and Grande colonies, and not “mixed” Petite colonies, but we comment on both for completeness.

Regarding “mixed” colony heteroplasmy in Figure 5:

There are two lines of evidence that point to these mixed-structure colonies containing heteroplasmic cells: (1) Reads containing tandem repeats are largely absent in these colonies. This points to rapid mtDNA recombination in these cells that opposes tandem repetition through rolling circle replication which is known to be the dominant mode of concatemer formation (Ling and Shibata, 2002). With such rapid recombination, which is known to generate transient heteroplasmy in most systems, it would be difficult for “mixed” structure cells to maintain homoplasmic states. We also note that this rapid recombination hypothesis is consistent with the generative model of mixed structures that was previously hypothesized in (Heyting et al., 1979). (2) Seven separate passages of progenitor colonies yield the same “mixed” structures. If instead of heteroplasmic cells we were actually seeing simply a mosaic of homoplasmic cells with diverse content, we would not expect this behaviour. Passaging in the case of a mosaic progenitor colony would result in segregation of these various structures across these passages.

Regarding Grande/Petite heteroplasmy in Figure 6:

In Grande colonies we leveraged fermentable/non-fermentable media to investigate heteroplasmic and homoplasmic contributions to sequenced mtDNA content. Here we make two arguments: (1) In Grande colonies under non-fermentable conditions (YPG), the presence of Petite mtDNA in bulk sequencing is likely to be contributed by heteroplasmic cells or recent heteroplasmy. This is because in contrast to the heteroplasmic cells containing Grande mtDNA, cells homoplasmic for Petite mtDNA cannot replicate under non-fermentable conditions. With this framework, any observed Petite mtDNA in YPG resides either in cells with Grande mtDNA in a heteroplasmic state, or is stuck in homoplasmic Petite cells which are unable to replicate but exist due to very recent heteroplasmy. (2) The increase in Petite mtDNA that accompanied a switch to fermentable media (YPD) is likely due to homoplasmic cells. Again, since under non-fermentable media homoplasmic Petites are suppressed, any increase in Petite mtDNA once we relax respiration requirements should primarily be due to homoplasmic Petite cells.

In Petite colonies where all cells regardless of mtDNA content are equally fit in fermentable media we make a different set of arguments. These arguments are based on the median value and variance of alternate structure abundance, which is highlighted in Figure 6b and 6c: (1) We argue that the median alternate structure frequency in Petites is lower than that of Grandes in Figure 6b due to stronger out-competition of alternate structures by primary structures in Petites. This is because small Petite primary structures replicate much faster than Grandes, and so alternate structures in Petites are less likely to take hold in this competition. Under this assumption, part of the alternate structure signals must be due to cells in at least transient heteroplasmic states to enable this competition. (2) Using Figure 6c and known properties of mtDNA transmission we argue that most of the variance in alternate structure frequencies is due to stochasticity in the generation of homoplasmic lineages. This is because mtDNA transmission bottlenecks and within-cell selection will favour the production of homoplasmic clones containing alternate structures. However, we also highlight that some colonies sequenced have distinguishably lower variance in alternate structure fraction (2 and 4b), which is at odds with this hypothesis. As such, these low variance alternate structure fractions point us towards heteroplasmic contributions in these particular colonies.

In response to the question about single-cell sequencing:

The first comment we make here is that part of what we hope is exciting about this analysis is that we are able to tease out heteroplasmic/homoplasmic contributions in the bulk sequencing of colonies, which is at the outset counterintuitive. While it is true that we provide indirect evidence for heteroplasmy, we make a set of conceptual arguments that are consistent with the theory of Petite mtDNA selection, and also provide experimental support for these ideas in media type switching in Grandes. It is also well established that heteroplasmy occurs during Petite generation, and the length of time heteroplasmy persists in yeast cells in the absence of selection (Ling and Shibata, 2004). The reviewer astutely suggests that single-cell sequencing would provide direct access to heteroplasmy. In fact, recent studies pioneering new single-cell sequencing approaches have revealed mtDNA heteroplasmy in humans cells (Maeda et al., 2020), albeit qualitatively, and a more quantitative description recently in (Lareau et al., 2021). We also note here that most commercial single-cell sequencing methods have been designed for mammalian cells. Many scRNA methods have just recently been adapted to and tested in yeast, such as in (Jariani et al., 2020), (Dohn et al., 2020), and (Urbonaite et al., 2021). With these recent developments we agree that single-cell sequencing would provide useful insights into mtDNA heteroplasmy in yeast like it has in humans, and is a wonderful direction for future work (commented on between Line # 745-750). To the best of our knowledge, a quantitative study of mtDNA heteroplasmy in yeast using these techniques also has not been performed. At the same time, however, the recent adaptation of these methods to yeast will be accompanied by a variety of complications that likely warrants a new study dedicated to this approach and its application to yeast mtDNA heteroplasmy.

Here we highlight an example of an interesting complication that may be encountered in single-cell sequencing of mtDNA in yeast that would require careful treatment. Unlike the bulk Nanopore sequencing we performed, single-cell sequencing requires significant DNA amplification prior to sequencing. Depending on the sequencing technology, this means one has to carefully consider both amplification bias and sequencing bias. For example, in PCR followed by Illumina sequencing, GC dense mtDNA is underrepresented, resulting in high variance in coverage (Aird et al., 2011). This GC dense mtDNA is also the limit that concatemers approach in excision cascades, which would make this a particularly insidious complication for our applications. The same coverage suppression occurs for AT rich sequences, which mtDNA in yeast is as a whole. In general, these types of biases would have to be carefully considered in attempts to quantify mtDNA heteroplasmy in yeast and most systems.

Discussion section regarding the outliers in repeat size – is it also possible that some outliers might simply reflect that selection is still in progress? Not all structures with sub-optimal fitness disappear instantly – selection often takes some time (ie replication cycles).

This is a very interesting question from the reviewer. One way that ongoing selection could manifest itself is in heteroplasmic cells that result in Grande colonies. For example, outlier mtDNA structures with similar fitness to wild-type mtDNA could in theory coexist for longer times. This would mean that our suppressivity measurements associated with these outlier structures, which only look at binary outcomes of colonies (Petite or Grande) derived from a heteroplasmic zygote, would label colonies composed of these heteroplasmic cells as Grandes. This could then explain why the suppressivity outliers we point to are lower than expected according to the model.

However, we note that in colonies composed of 10^6^ cells after ~20 generations of growth, even in the absence of differential selection most cells are homoplasmic (Ling and Shibata, 2004). In addition, the fate of the colony (Grande vs Petite) is driven by the first few cells derived from the zygotes. More quantitatively, a petite segregant on Nth division impacts (½)^N^ fraction of the colony. Taken together, our suppressivity measurements are ensemble averages in Grande/Petite mtDNA competition outcomes in the early progeny of zygotes.

This means, as the reviewer suggests, that the short timescale of competition may be noisy and not allow deterministic outcomes for every mating event. However, across hundreds of mating events that encompass thousands of replications during competition, even weak selection should be detectable. For example, even though at early times a coexisting structure with a marginal fitness advantage won’t always out-compete wild-type cells, in an ensemble average of outcomes (across thousands of replication events in hundreds cells) suppressivity should be still elevated above 50%.

Discussion, last paragraph: I do not agree with the conclusion that these findings suggest that feral yeasts have maintained their mtDNA as a result of continuous selection for respiratory metabolism. It would be useful to consider and discuss the fact that feral yeasts generally have much lower petite formation; and consider if and how the use of a laboratory strain like W303 influences the results.Note that laboratory strains and some beer yeasts likely lost the mechanisms that reduce mt DNA instability because they have been relieved from selection for respiratory metabolism for many years of repeated growth under fermentative conditions in laboratory medium (high glucose, inducing Crabtree repression); but feral yeasts do not need continuous selection for respiration to maintain their mtDNA.

We thank the reviewer for this important comment. It is true that laboratory strains generally have higher Petite frequencies than feral yeasts, and numerous mutations in lab strains have been shown to increase Petite frequencies compared to feral yeasts (Dimitrov et al., 2009). As the reviewer suggests, this can largely be attributed to the lack of selection for respiring cells historically during brewing, or in laboratory conditions that allowed the mtDNA instability we observe to flourish. By the same token, this indicates that strong selection for respiration is central to why we see low Petite frequencies in feral yeasts.

The reviewer brings up the important point that mtDNA maintenance mechanisms may have been lost during this relaxation of selection for respiration. But this doesn’t mean mtDNA fragmentation is exclusively enabled by these lab strain mutations. In fact, the repetitive nature of the mt genome primarily responsible for excisions is also conserved in many feral yeasts, which likely means fragmentation also occurs in feral strains but just at a lower rate. Regarding the comment on continuous selection for respiration, we maintain that over evolutionary timescales in Petite positive yeasts, respiration selection is indeed what maintains intact mtDNA. Fragmented mtDNA incurs a negative cost due to unproductive replication and transcription under fermentation conditions. Over short timescales though, as the reviewer suggests, continuous selection for respiration is not necessary to maintain intact mtDNA, especially if mtDNA fragmentation occurs at a low rate.

The point we were trying to make in the last paragraph of the discussion is that given the repetitive nature of the genome, which is known to introduce instability, population level selection for respiration is likely a part of what enabled the evolution of this repetitive structure. This is especially true if mtDNA instability, or the repetitiveness that causes it, serves other purposes, and population level selection allows increased mtDNA instability without detriment at the population level. We have rewritten this paragraph to emphasize these points more clearly, and think it serves as an intriguing hook for readers to think about the levels of selection at play in this system (Line # 814 – 830).

Reviewer #2 (Recommendations for the authors):1. In Figure 2b and 2c, having the entire schematic of the yeast mitochondrial genomes rather than just sub-sections would make your point about the repeat concatemer structure found Petite genomes while also making the work more accessible by those who don't use yeast as their model organism.

We thank the reviewer for this suggestion. We have rearranged Figure 2, merging panels 2b and 2c into one and included the entire schematic of the yeast mitochondrial genome in this new merged panel. As the reviewer suggests this makes it much more clear to an audience unfamiliar with yeast mtDNA where the genome features in the sequencing reads come from. It also immediately highlights that Petite reads are concatemers with subgenomic repeat units (Line #186 and Line #195-202).

2. Additionally, please move the legend in figure 2c off the figure. Also, increasing the visibility of the red breakpoint locations (especially in the Grande genome illustration) will allow readers to better compare the structural differences in the genomes.

We thank the reviewer for this suggestion. Both changes have been implemented in Figure 2 (Line #186).

3. The use of the x,y plot for figure 2d is a bit confusing. Mapping all the breakpoint locations to schematic of the yeast mtDNA genome might more clearly illustrate the greater quantity and distribution of breakpoints found in the Petite genomes as compared to the Grande genomes.

We thank the reviewer for this suggestion. However, we note that to illustrate where breakpoint locations are we need two positions on the mtDNA genome because breakpoints represent two disjoint locations being brought together. We believe this is best illustrated in a 2D linear-axis plot, as this shows with a single marker where each breakpoint is located without the need for curves connecting different portions of the genome. We have added another sentence to the figure description to make this plot more clear, describing that the X and Y coordinates of markers represent the disjoint locations in mtDNA that interact to produce the breakpoint (Line #205). We have also added an alternate circular version of this plot in Appendix 1 – Figure 2, which separates non-inverted and inverted junction plots into two panels and uses curves to represent breakpoints if this is more appealing to some readers (referenced in Line #213).

4. In the Discussion section, citing other systems that this sequencing and structural inference system could be applied to would help to increase the relevance of this system outside of your specific research question.

We thank the reviewer for this comment. We now highlight in the discussion (Line # 832-857) that our methodology lends itself to studying systems like plants where complex and repetitive mtDNA coexists. We also show at the end of the discussion that there are numerous parallels between Petite formation and mtDNA deletions in humans.

Reviewer #3 (Recommendations for the authors):It is imperative that all data is made available to the reader in the raw and processed format, with annotation sufficient to repeat analyses and to run novel analyses. At this point I do not see clear data availability section.

We thank the reviewer for this comment. A data and code availability section has been added to the manuscript with a DOI for both code and raw data that will be activated upon publication. A reviewer URL to access all data and code is also attached to this submission. Additionally we have included processed data, which is the output of the structural detection pipeline acting on raw data, alongside Python scripts that use this processed data to generate all figures in the manuscript. The processed data and code is available from: https://github.com/javathejhut/ContingencyAndSelection (Line #1164-1169). We have also added a description of how we addressed marginal cases in breakpoint clustering in the published code to the methods section and Appendix 2, which we noticed was missing in a code review in the response to this comment (Line #996-1001).

References:

Lareau, C. A., Ludwig, L. S., Muus, C., Gohil, S. H., Zhao, T., Chiang, Z., Pelka, K., Verboon, J. M., Luo, W., Christian, E., Rosebrock, D., Getz, G., Boland, G. M., Chen, F., Buenrostro, J. D., Hacohen, N., Wu, C. J., Aryee, M. J., Regev, A., and Sankaran, V. G. (2021). Massively parallel single-cell mitochondrial DNA genotyping and chromatin profiling. *Nature biotechnology*, *39*(4), 451–461. https://doi.org/10.1038/s41587-020-0645-6

Maeda, R., Kami, D., Maeda, H. *et al.* High throughput single cell analysis of mitochondrial heteroplasmy in mitochondrial diseases. *Sci Rep* 10, 10821 (2020). https://doi.org/10.1038/s41598-020-67686-z

Dohn, R.; Xie, B.; Back, R.; Selewa, A.; Eckart, H.; Rao, R.P.; Basu, A. mDrop-Seq: Massively Parallel Single-Cell RNA-Seq of *Saccharomyces cerevisiae* and *Candida albicans*. Vaccines 2022, 10, 30. https://doi.org/ 10.3390/vaccines10010030

Jariani, A., Vermeersch, L., Cerulus, B., Perez-Samper, G., Voordeckers, K., Van Brussel, T., Thienpont, B., Lambrechts, D., and Verstrepen, K. J. (2020). A new protocol for single-cell RNA-seq reveals stochastic gene expression during lag phase in budding yeast. *eLife*, *9*, e55320. https://doi.org/10.7554/*eLife*.55320

Urbonaite, G., Lee, J.T.H., Liu, P. *et al.* A yeast-optimized single-cell transcriptomics platform elucidates how mycophenolic acid and guanine alter global mRNA levels. *Commun Biol* 4, 822 (2021). https://doi.org/10.1038/s42003-021-02320-w

Aird, D., Ross, M.G., Chen, WS. *et al.* Analyzing and minimizing PCR amplification bias in Illumina sequencing libraries. *Genome Biol* 12, R18 (2011). https://doi.org/10.1186/gb-2011-12-2-r18

Hawkins, M., Retkute, R., Müller, C. A., Saner, N., Tanaka, T. U., de Moura, A. P. S., and Nieduszynski, C. A. (2013). High-Resolution Replication Profiles Define the Stochastic Nature of Genome Replication Initiation and Termination. *Cell Reports*, *5*(4), 1132–1141. doi:10.1016/j.celrep.2013.10.014